

# Past and future response of Greenland's tidewater glaciers to submarine melting

Donald Slater[1], Fiamma Straneo[1], Denis Felikson[2], Chris Little[3], Heiko Goelzer[4,5], Xavier Fettweis[6], and James Holte[1]

[1]Scripps Institution of Oceanography, University of California San Diego, La Jolla, CA, USA
[2]NASA Goddard Space Flight Center, Greenbelt, MD, USA
[3]Atmospheric and Environmental Research, Inc., Lexington, MA, USA
[4]Utrecht University, Institute for Marine and Atmospheric Research, Utrecht, the Netherlands
[5]Laboratoire de Glaciologie, Université Libre de Bruxelles, Brussels, Belgium
[6]Laboratory of Climatology, Department of Geography, University of Liège, Liège, Belgium

**Correspondence:** Donald Slater (daslater@ucsd.edu)

**Abstract.** The effect of the North Atlantic Ocean on the Greenland Ice Sheet through submarine melting of Greenland's tidewater glacier calving fronts is thought to be a key driver of widespread glacier retreat, dynamic mass loss and sea level contribution from the ice sheet. Despite its critical importance, problems of process complexity and scale hinder efforts to represent the influence of submarine melting in ice sheet-scale models. Here we propose parameterizing tidewater glacier

terminus position as a simple linear function of submarine melting, with submarine melting in turn estimated as a function of subglacial runoff and ocean temperature. The relationship is tested, calibrated and validated using datasets of terminus position, runoff and ocean temperature covering the full ice sheet and surrounding ocean from the period 1960-present. We demonstrate a statistically significant link between multi-decadal tidewater glacier terminus position and submarine melting and show that the proposed parameterisation has predictive power when considering a population of glaciers. An illustrative

21st century projection is considered suggesting that tidewater glaciers in Greenland will undergo little further retreat in a low emissions RCP2.6 scenario. In contrast, a high emissions RCP8.5 scenario results in a median retreat of ∼6 km, with 35% of glaciers experiencing retreat exceeding 10 km. Our study provides a long-term and ice sheet-wide assessment of the sensitivity of tidewater glaciers to submarine melting and proposes a practical and empirically validated means of incorporating ocean forcing into models of the Greenland ice sheet.

## 1  Introduction

Discharge of ice from marine-terminating glaciers around the margin of the Greenland Ice Sheet is responsible for 9.1 mm of Greenland's 1972-2018 total sea level contribution of 13.7 mm (Mouginot et al., 2019) and, together with increased surface melting, has resulted in Greenland becoming the fastest growing contributor to global sea level (Chen et al., 2017). Increased discharge from tidewater glaciers is understood to be a response to a warming of the ocean and fjords surrounding the ice sheet

that, in concert with increased surface melting and subglacial runoff, has resulted in increased submarine melting and calving at tidewater glacier termini (Straneo and Heimbach, 2013). Given projections of continued atmospheric and oceanic warming



in Greenland (Yin et al., 2011; Fettweis et al., 2013), it is clear that capturing the described chain of ocean and ice dynamic processes in models is a basic requirement if we are to produce accurate sea level projections.

Considering first the ocean processes, the Greenland ice sheet interacts directly with the ocean at around 300 tidewater glacier calving fronts, several kilometers wide and several hundred metres deep (Rignot and Mouginot, 2012). A handful of

floating ice tongues/shelves persist in Northern Greenland (Wilson et al., 2017) and may periodically form at larger glaciers further south (Kehrl et al., 2017). Ocean heat to drive submarine melting of these glaciers is readily available due to the presence of warm subtropical waters around Greenland (Straneo et al., 2012). To reach calving fronts these waters must first cross the continental shelf, a passage which may be promoted by cross-shelf troughs (Fraser et al., 2018), and then travel up fjords, a passage which may be promoted by fjord circulation but impeded by the presence of sills (Motyka et al., 2003; Straneo

et al., 2011; Jackson et al., 2014; Carroll et al., 2017). Once at calving fronts, these waters may be entrained into vigorous plumes initiated by subglacial runoff driving rapid melting during the summer (Jenkins, 2011; Mankoff et al., 2016). Away from plumes or outside of the summer season, submarine melting may be driven by wider fjord circulation (Slater et al., 2018), or by self-sustained convection (Magorrian and Wells, 2016).

Many of these ocean processes have been captured by models, yet accurately representing plume dynamics and calving

front circulation requires resolution on the order of 10 m (Xu et al., 2012), hence such models are limited to the heads of individual fjords. Through parameterisation of plume dynamics (Cowton et al., 2015) it is possible to run regional ocean models of large fjords and the adjacent continental shelf (Cowton et al., 2016; Fraser et al., 2018), but with resolution on the order of 500 m it remains prohibitively expensive to extend such models to the full ice sheet and surrounding ocean basins. Furthermore, predictions of submarine melt rates from such process-based models have large uncertainties as we still lack

reliable observations of submarine melt rates from tidewater glaciers in Greenland (Straneo and Cenedese, 2015; Jackson and Straneo, 2016). As such it remains challenging, especially at an ice sheet scale, to model the ocean processes necessary for accurate process-based projection of sea level.

Turning to glacier frontal ice dynamics, calving of solid ice may occur through various styles, processes, and magnitude of event (Benn et al., 2007; How et al., 2019). Some of these processes may respond to submarine melting. For example,

focused melting can incise undercut chimneys into calving fronts (Fried et al., 2015; Rignot et al., 2015), which may drive small calving events from ice which has been undermined or large calving events through altering the stress distribution further up-glacier (O'Leary and Christoffersen, 2013; Benn et al., 2017a; Ma and Bassis, 2019). Other processes imply that calving responds primarily to the atmosphere, for example hydrofracture driven by the presence of water in crevasses (Benn et al., 2007). Yet others may be driven primarily by ice dynamics and bed topography, for example the advection of ice into deep

water, resulting in a buoyant torque on the terminus (James et al., 2014; Wagner et al., 2016). There is also increasing evidence for the important role played at some glaciers by ice mélange (Amundson et al., 2010; Moon et al., 2015; Robel, 2017), which acts to inhibit calving by producing a backstress on the terminus.

High-resolution models of individual glaciers show promise of capturing calving processes (Åström et al., 2014; Benn et al., 2017a; Todd et al., 2018; Bondzio et al., 2016), and have even been run at a regional scale (Morlighem et al., 2019), but require

resolutions of around 100 m or less. This is roughly an order of magnitude finer than the current generation of ice sheet-scale



models (Goelzer et al., 2018). Thus it remains extremely challenging to simulate the entire Greenland ice sheet with sufficient detail to resolve individual glaciers and their calving processes. Parameterisation of calving processes has furthermore proven difficult due to the diversity of styles and difficulty of collecting the relevant datasets, and there is currently no calving law which has been extensively validated at tidewater glaciers (Benn et al., 2017b).

For these reasons, which might be summarised as problems of process understanding and scale, inclusion of ice sheet-ocean processes in Greenland Ice Sheet models has proven difficult. A number of ad-hoc methods have therefore been used (Price et al., 2011; Goelzer et al., 2013; Nick et al., 2013; Fürst et al., 2015; Golledge et al., 2019), but such approaches often focus on large glaciers and rely on scaling arguments to obtain full ice sheet response, and/or are not faithful to the processes now believed to be responsible for terminus position change. There is therefore an urgent need for methods of modeling the influence

of the ocean on the Greenland ice sheet which satisfy the somewhat competing requirements of process fidelity and practical scalability.

     To motivate such a method, submarine melting has emerged as the leading forcing amongst the processes described driving tidewater glacier retreat (Straneo and Heimbach, 2013; Luckman et al., 2015; Benn et al., 2017b). Submarine melt rates are likely too small to account for all of the observed retreat in most locations; it is instead thought that increased submarine

melting initiates a dynamic response involving increased calving and glacier acceleration and retreat (Morlighem et al., 2016). The potential dynamic response is, however, understood to be highly sensitive to topography. Bed topography that shallows or deepens inland is thought to promote stability and retreat respectively (Schoof, 2007; Catania et al., 2018). Similarly, it is thought that fjords that narrow or widen inland promote stability and retreat respectively (Enderlin et al., 2013; Carr et al., 2013). Thus topography lends a large degree of individuality to glacier response to climate forcing, potentially obscuring a

simple relationship between terminus position and submarine melting (Murray et al., 2015; Carr et al., 2017).

     Conversely, there is a degree of commonality in observed tidewater glacier behaviour. For example, the recent acceleration and retreat of tidewater glaciers is widespread; Murray et al. (2015) show that 94% of Greenland's significant tidewater glaciers retreated between 2000-2010. The onset and evolution of the recent response is also similar within regions (Rignot and Kanagaratnam, 2006; Moon et al., 2012; Khan et al., 2010; Catania et al., 2018). Porter et al. (2018) build on such commonality

to find a significant correlation between glacier dynamic thinning and nearby ocean heat content for all glaciers in Greenland except those in the west. Jensen et al. (2016) find significant regional correlations of tidewater glacier area change with various climate indices such as sea surface temperature, sea ice concentration and North Atlantic Oscillation index. Cowton et al. (2018) showed that between 1993 and 2012, combined atmospheric and oceanic variability explained 54% of change in terminus position across 10 tidewater glaciers in east Greenland. Thus while individual glacier response is heterogeneous,

more homogeneous behaviour may emerge as groupings of glaciers are considered over larger spatial scales and longer time scales, lending promise to simple parameterisations.

     Motivated by the urgent need for an ice sheet-ocean coupling approach that respects the key processes but scales to practical applications, and given the leading role of submarine melting in recent tidewater glacier retreat and the commonality of this response, we here propose expressing tidewater glacier retreat as a simple linear function of estimated submarine melt rate. We

use the largest assembled dataset to date of past terminus positions and climate to demonstrate the existence of a statistically



significant relationship between terminus position and submarine melt at the ice sheet scale, and to calibrate and validate the retreat parameterisation. We apply the parameterisation to generate 21st century retreat projections driven by climate forcing from a single climate model. The resulting parameterisation is the standard approach which has been recommended to ice sheet modelers taking part in the Ice Sheet Model Intercomparison Project (ISMIP6, Nowicki et al., 2016), which aims to produce

sea level projections for Greenland for the coming 6th Assessment Report of the Intergovernmental Panel on Climate Change (IPCC AR6).

## 2   Methods

### 2.1   Retreat parameterisation

We draw on detailed modeling of submarine melting at tidewater glacier calving fronts, together with the observation that

tidewater glacier retreat is most frequent in the summer when plumes are active, to suggest estimating submarine melting at each glacier by $Q^{0.4}\,\mathrm{TF}$, where $Q$ is the summer (June-July-August) mean subglacial runoff. TF is the ocean thermal forcing (the temperature of ocean waters above the in-situ freezing point), typically considered homogeneous across an individual calving front and sampled at the grounding line depth or averaged over the deeper part of the water column (Jenkins, 2011; Xu et al., 2013; Sciascia et al., 2013; Cowton et al., 2015; Slater et al., 2016; Rignot et al., 2016). The inclusion of runoff

$Q$ represents the process understanding that plumes become more vigorous and drive more submarine melt as subglacial runoff increases. The thermal forcing TF represents the ocean heat available to drive melting. Together with the motivation in the introduction, we here propose that tidewater glacier retreat $\Delta L$ be parameterised as a linear function of the change in submarine melting $\Delta\left(Q^{0.4}\,\mathrm{TF}\right)$

$$\Delta L = \kappa\,\Delta\left(Q^{0.4}\,\mathrm{TF}\right) \tag{1}$$

The approach is similar to that proposed in Cowton et al. (2018), which suggesting parameterizing change in terminus position as $dL/dt \propto d/dt\left(Q\,\mathrm{TF}\right)$. The present study builds on their results by significantly expanding the temporal and spatial calibration and validation of the retreat parameterisation, by quantifying the uncertainty associated with the parameterisation, and by providing forward projections.

     We develop and test our parameterisation as follows. First, past observations or reconstructions of terminus positions, runoff

and ocean temperatures are used to validate and calibrate the parameterisation (Fig. 1). These observations span the time period 1960-present; while we do have terminus position records from well before 1960, the subsurface ocean data coverage pre-1960 is very limited, so that we considered our estimates of ocean thermal forcing to be reliable only post-1960. In order to reduce the temporal bias in the dataset (i.e., more recent years having more terminus positions), and to bring all datasets onto a common time axis, throughout this paper we bin the terminus positions, runoff and ocean thermal forcing into 5-year time periods. This

also acts as a form of low-pass filter, removing short-term variability which is not important to the longer-term trends which we aim to capture. Following this exercise, which provides a probabilistic range of values of the sensitivity parameter $\kappa$, future



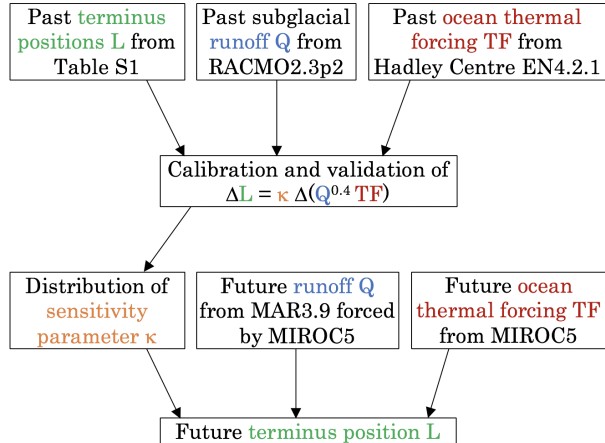

**Figure 1.** Schematic of approach taken in this study. We use observations/reconstructions of past terminus position, subglacial runoff and ocean temperature to calibrate and validate a simple parameterisation for tidewater glacier retreat. We use the resulting parameterisation, together with projections of future subglacial runoff and ocean temperature, to project future terminus position.

glacier retreat can be projected with the use of climate model output to estimate the right hand side of the parameterisation (Fig. 1).

## 2.2 Past observations

### 2.2.1 Terminus positions

5 Terminus positions are taken from a number of published sources (Fig. 2, Table S1, Andresen et al., 2012; Steiger et al., 2018; Lea et al., 2014; Haubner et al., 2018; Catania et al., 2018; Cowton et al., 2018; Moon and Joughin, 2008; Joughin et al.; Bunce et al., 2018; Carr et al., 2017). Broadly, these records can be organised into long records from a handful of individual glaciers, and shorter satellite-era records from almost all significant tidewater glaciers in Greenland (Figs 2a and 2b). Thus the number of records from before 1992 is rather limited, while coverage from the year 2000 is nearly complete (Fig. 2c). In total the

10 dataset includes 191 of the 211 fastest flowing tidewater glaciers in Greenland identified by Rignot and Mouginot (2012). We believe that the records we have collated constitute the most complete such dataset to date. We removed 3 glaciers known to have persistent ice shelves (Peterman, Ryder and 79N) as the dynamics of ice shelf fronts differ from tidewater glaciers with approximately vertical termini. Processing and merging of the terminus positions into a common dataset is described in the supporting information.

### 15 2.2.2 Runoff

Runoff for each of the 191 tidewater glaciers is estimated using the regional climate model RACMO (Noël et al., 2018). The runoff dataset is statistically downscaled to 1 km from the output of RACMO2.3p2 at 5.5 km horizontal resolution. Compared

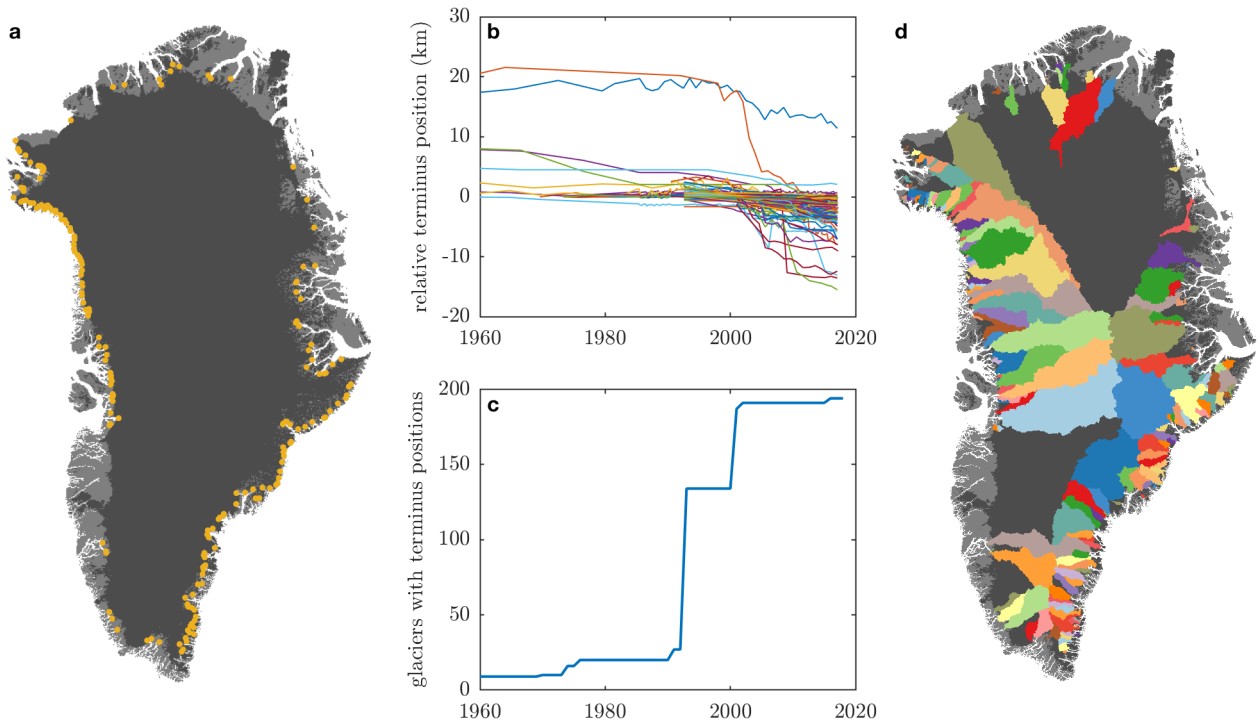

**Figure 2.** (a) Locations of tidewater glaciers considered, (b) overview of terminus position change at all glaciers, (c) number of glaciers in the dataset as a function of time, and (d) hydrological drainage basins for each of the 194 tidewater glaciers in the terminus position dataset, calculated using hydrological flow routing with topography from BedMachine version 3 (Morlighem et al., 2017).

to the data discussed in Noël et al. (2018), no model physics have been changed. Refined horizontal resolution of the host model, i.e., 5.5 km instead of 11 km, better resolves gradients in SMB components at the ice sheet margins. The simulation is forced at its boundaries by ERA-40 and ERA-Interim and spans the full time period 1960-present considered here.

Runoff is routed to the ice sheet margins based on the hydrological potential (Shreve, 1972; Schwanghart and Scherler, 2014). We take surface and bed topography from BedMachine version 3 (Morlighem et al., 2017; Howat et al., 2014) and assume that subglacial water pressure is equal to ice overburden pressure. This process defines a hydrological basin for each tidewater glacier (Fig. 2d), over which runoff from the regional climate model is summed to give an estimated runoff for the tidewater glacier. We assume that the drainage basin is fixed in time, and that the runoff is transferred instantaneously from melting at the surface to emerging at the grounding line. Given the strong seasonality in runoff, and since most tidewater glacier retreat is observed to take place during the summer (Fried et al., 2018), we consider the mean summer runoff over June, July and August. An analysis of the relationship between annual and summer runoff (not shown) indicates that no significant differences in results or projections would arise from using annual rather than summer runoff.





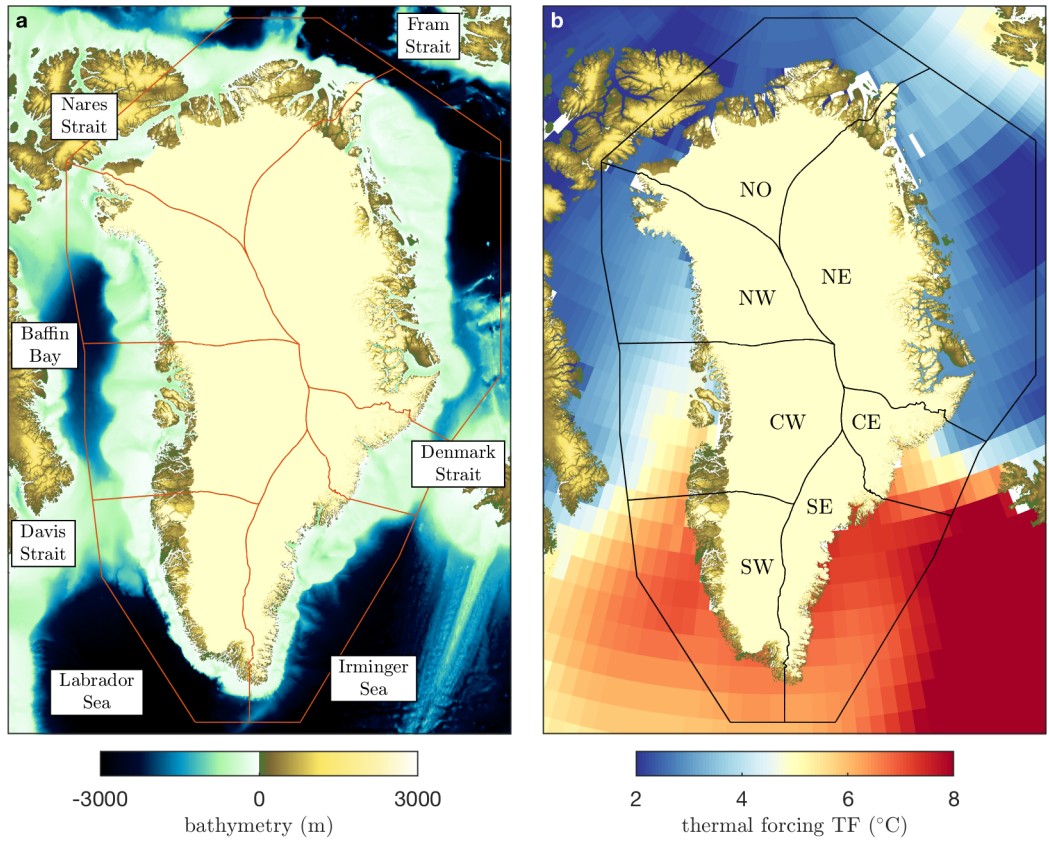

**Figure 3.** (a) ocean bathymetry around Greenland and (b) 200-500 m depth-average, 1995-2014 time-average thermal forcing TF around Greenland from the EN4 dataset (Good et al., 2013). Red lines on (a) and black lines on (b) show the ice-ocean sectors over which the thermal forcing is averaged. The location of key geographic features mentioned in the text are shown on (a) while the ice-ocean sectors are labelled on (b).

### 2.2.3 Ocean temperature

Subsurface ocean temperatures around Greenland (Fig. 3) come from the Hadley Centre EN4 dataset version 4.2.1 (Good et al., 2013). We use the EN4 monthly objective analyses, which is a 1 degree latitude by 1 degree longitude gridded product of temperature and salinity formed from oceanographic profiles. The dataset covers the period 1900 to present at monthly intervals, though profile data is limited in the early part of the time series and in the northern half of Greenland, so that as already described we only consider the time period 1960-present. We calculate gridded depth-variable ocean thermal forcing as $T(z) - (\lambda_1 S(z) + \lambda_2 + \lambda_3 z)$, where $T(z)$ is potential temperature, $S(z)$ is salinity, $z$ is depth, and $\lambda$ values are as in Jenkins (2011). An overview of the spatial and temporal coverage of profile data going into the gridded product is shown in Figs. S3 & S4.



The ocean forcing felt by tidewater glaciers is related to, but not the same as, the ocean forcing available on the continental shelf and in the ocean basins around Greenland. In particular, offshore warm deep waters and their seasonal cycle are modified by fjord circulation processes before reaching calving fronts (Straneo et al., 2011; Mortensen et al., 2011; Jackson et al., 2014; Gladish et al., 2015b). While rapid progress has been made in understanding fjord circulation and in mapping fjord bathymetry,

we as yet lack the simple parameterisations and complete datasets required to include fjord effects in this study. As such, we here depth-average the thermal forcing between 200 and 500 m, this being the characteristic grounding line depth range of Greenlandic tidewater glaciers (Morlighem et al., 2017). We also average over each year and spatially average over defined sectors (Fig. 3) to give a thermal forcing TF per year and per sector to be used as a proxy for the thermal forcing felt by tidewater glaciers. The use of sector averages means that every glacier in a given sector experiences the same thermal forcing,

and is justified at present by the sparsity of oceanographic data available around Greenland and our inability to account for the effect of individual fjords.

Sectors are chosen as a compromise between oceanic basins (Fig. 3a), ocean temperature gradients (Fig. 3b), and ice sheet drainage basins, where the boundaries are similar to previous studies (Shepherd et al., 2012; Mouginot et al., 2019). We thus have sector boundaries over Davis Strait, Nares Strait and Fram Strait, and we separate the Irminger Sea from the Labrador

Sea with a boundary close to Cape Farewell (Fig. 3a). We separate west Greenland from north-west Greenland due to the large meridional temperature gradient in Baffin Bay (Fig. 3b). Finally, we create a small central-east Greenland sector which includes the whole Denmark Strait region; from the ocean perspective it would be desirable to instead place a sector boundary on the Denmark Strait, but extending this boundary onto the ice sheet is awkward due to the presence of Kangerdlugssuaq Glacier. We extended the ocean sectors beyond the continental shelf towards the centre of the ocean basins because oceanographic data

coverage improves significantly beyond the shelf (Figs. S3 & S4); we note however that the thermal forcing obtained is not sensitive to the exact definition of this boundary.

### 2.3   Future climate forcing

To generate projections of 21st century terminus position, estimates of future subglacial runoff and ocean thermal forcing are required (Fig. 1). These are estimated using the global climate model MIROC5 (Watanabe et al., 2010) under a low (RCP2.6)

and a high (RCP8.5) greenhouse gas emissions scenario. We emphasise that these projections are intended as an illustration for a single model rather than a rigorous result, as global climate models can differ significantly in their projected ocean and atmospheric warming (e.g. Yin et al., 2011).

Ice sheet surface mass balance is generally included in only a basic manner in global climate models, including MIROC5, and we therefore estimate future subglacial runoff $Q$ using 1950-2100 simulations in the regional climate model MAR, forced

at its boundaries by MIROC5 (Fettweis et al., 2013). With respect to MAR simulations performed in Fettweis et al. (2013), the latest version of MAR (3.9.6) is used here at a resolution of 15 km. The outputs were furthermore downscaled to 1 km to better account for subgrid-scale topography (Franco et al., 2012; Howat et al., 2014). Finally it should be noted that the MAR simulations use a fixed present day topography which is acceptable to 2100 according to Le clec'h et al. (2019). Surface melting from MAR is summed over each of the tidewater glacier drainage basins (Fig. 2d) to give a subglacial runoff time



series for each glacier extending to 2100. Future thermal forcing TF is obtained directly from MIROC5 output by following the same procedure as for the observations: we convert potential temperature to thermal forcing, we average the model output over each year, over the depth range 200-500 m, and spatially over the ice-ocean sectors (Fig. 3). The time series are bias corrected per glacier (for runoff) and per sector (for ocean thermal forcing) to ensure they are consistent with observations in the present

day, ensuring the transition from past to future climate forcing is continuous (Appendix A).

### 2.4   Statistics

We assess the statistical significance of relationships between 5-year binned terminus position and parameterised submarine melt rate as follows. Since trends exist in both time series which could lead to spurious correlation, the time series are first detrended by subtracting a linear trend in time over the full length of the dataset (Santer et al., 2000; Hanna et al., 2013).

Linear regression is then performed on the detrended time series to obtain a linear coefficient $b$ and standard error $s_b$. We test whether $b$ is significantly different from 0 by forming the ratio $\tau_b = b/s_b$ and performing a two-sided t-test on $\tau_b$ with $N$ degrees of freedom. To account for temporal autocorrelation in the time series, we reduce the degrees of freedom by defining $N = n(1 - r_1 r_2)/(1 + r_1 r_2)$, where $n$ is the number of values in the time series, and $r_1$ and $r_2$ are the lag one autocorrelation coefficients for the terminus position and submarine melt time series (Santer et al., 2000; Hanna et al., 2013). If, following this

procedure, we find $b$ to be significantly different from 0 (say at the 5% level), this implies a significant relationship between terminus position and submarine melt. In this study we apply this procedure to assess significance at both an individual glacier and Greenland-wide level.

Note that while we assess statistical significance using detrended time series, we assess the sensitivity $\kappa$ of terminus position to submarine melt using the original (trended) time series. We consider this necessary because we wish to ensure that the

parameterisation Eq. (1) captures past behavior of tidewater glaciers in Greenland as closely as possible, and because there is strong process evidence linking increased submarine melting to tidewater glacier retreat. Thus values of $\kappa$ and correlation coefficients $R^2$ are calculated using the original time series as we wish to give an indication of how well Eq. (1) captures past behaviour, but values of statistical significance $p$ are calculated using detrended time series and reduced degrees of freedom, as described above.

## 3   Results

### 3.1   Relationship between submarine melting and terminus position

We begin our analysis by examining two glaciers, Kangiata Nunata Sermia in SW Greenland and Store Glacier in CW Greenland, which exemplify the diversity of glacier response to submarine melting (Fig. 4). The terminus of Kangiata Nunata Sermia was stable within ±500 m from 1960 until 2000, before undergoing a rapid retreat of 2 km followed by restabilisation (Fig. 4a).

Subglacial runoff has increased steadily since 1980 (Fig. 4b) while SW Greenland ocean temperatures have warmed since 1995 after approximately 20 years of colder conditions (Fig. 4c). Submarine melting (Fig. 4d), combining runoff and thermal forcing,



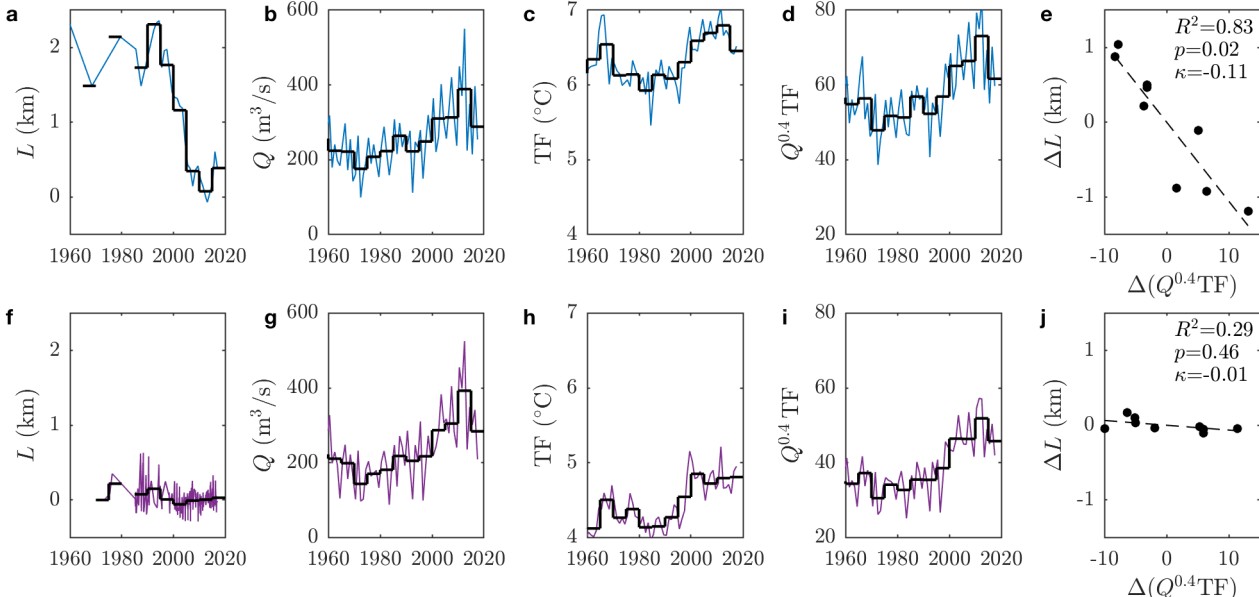

**Figure 4.** Example calibration of the retreat parameterisation for two glaciers: Kangiata Nunata Sermia in SW Greenland (top row), and Store glacier in CW Greenland (bottom row). (a, f) terminus position, (b, g) summer runoff, (c, h) ocean thermal forcing and (d, i) parameterised submarine melting. Light lines show all values in the datasets while the heavy black lines show 5-year binning. (e, j): 5-year binned values of parameterised submarine melt anomaly (x-axis) versus terminus position anomaly (y-axis). Anomalies are calculated as the difference from the mean of all values. Text on (e, j) shows the correlation coefficient ($R^2$), the significance of the regression ($p$) and the linear coefficient ($\kappa$).

was stable from 1960 to 1995 before a rapid increase until 2010 and a small decrease since. There is a statistically significant relationship between terminus position and submarine melt rate ($p = 0.02$) with variability in submarine melting explaining 83% of terminus position change and a sensitivity coefficient $\kappa = -0.11$ (Fig. 4e).

Store Glacier has in contrast remained stable since at least 1970, with a very moderate retreat of a few hundred metres in the
5   1990s (Fig. 4f). Subglacial runoff has also increased steadily until the past few years (Fig. 4g) and ocean temperatures in CW Greenland show an increasing trend throughout most of the period (Fig. 4h). Estimated variability in submarine melting (Fig. 4i) explains only 29% of terminus position change (Fig. 4j). The estimated sensitivity coefficient is $\kappa = -0.01$, suggesting that Store Glacier is relatively insensitive to submarine melting, a conclusion previously attributed to the particular bed topography of the glacier (Morlighem et al., 2016; Todd et al., 2018). Unsurprisingly therefore, the relationship between submarine melting
10   and terminus position is not found to be significant at Store Glacier.

This procedure can be repeated for every glacier around the ice sheet (Fig. 5). The sensitivity coefficient $\kappa$ relating submarine melt forcing to change in terminus position forms a skewed distribution with very few positive $\kappa$ values (increased submarine melting associated with glacier advance) and a long tail of negative $\kappa$ values (Fig. 5a). The sharply-peaked distribution suggests that many glaciers in Greenland show similar order-of-magnitude sensitivity to submarine melting. The median value is $\kappa_{50} =$



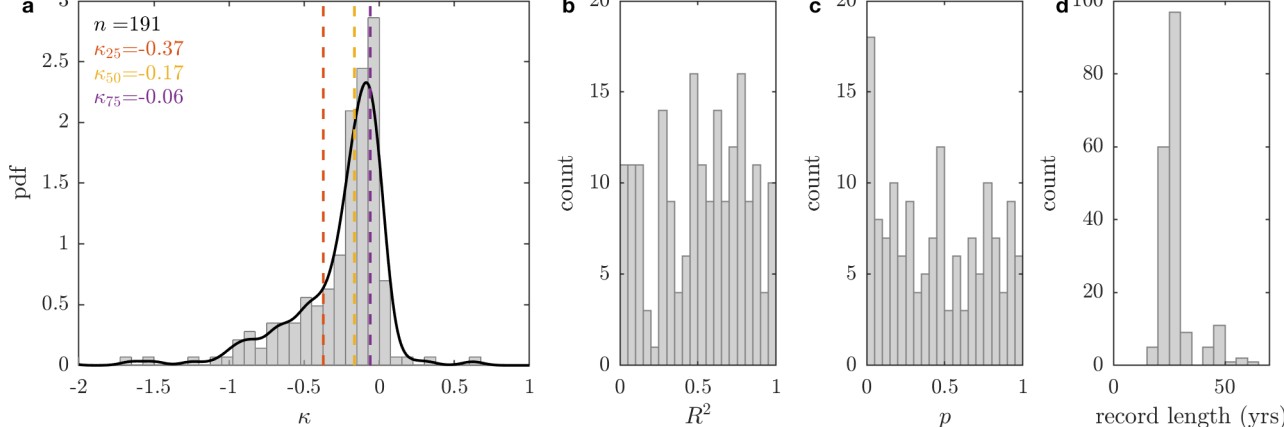

**Figure 5.** Result of linear regression (as in Figs. 4e and 4j) for all glaciers. Histograms of (a) sensitivity coefficient $\kappa$, (b) correlation coefficient $R^2$, (c) statistical significance $p$, and (d) length of record entering the regression. Vertical dashed lines on (a) indicate the quartiles of the distribution, while the solid black line shows a kernel distribution fit to the histogram. Glaciers with a record less than 15 years were excluded from all plots.

$-0.17$, while the lower and upper quartiles take values $\kappa_{25} = -0.37$ and $\kappa_{75} = -0.06$ respectively. Kangiata Nunata Sermia with $\kappa = -0.11$ (Fig. 4e) therefore shows fairly average sensitivity to submarine melting, while Store Glacier with $\kappa = -0.01$ (Fig. 4j) is more insensitive than 90% of glaciers in Greenland. Variability in submarine melting explains greater than 50% of terminus position change at 105 glaciers (Fig. 5b), but the relationship is statistically significant at the 5% level at only 18

glaciers (Fig. 5c). Finally, although we do have several glaciers for which the regression is performed on a record longer than 50 years, for the majority of glaciers the length of record is less than 30 years (Fig. 5d). This results from a combination of lack of terminus positions before the satellite era and the sparsity of ocean data until the past few decades.

We examine sector-to-sector variability in sensitivity to submarine melting by conducting the same analysis for each sector separately (Fig. 6). The five more southerly and easterly sectors (SE, SW, CE, CW and NE, see Fig. 3b) show remarkably similar

sensitivity of terminus position to submarine melting, as indicated by similar distributions for $\kappa$ (Fig. 6a). The most northerly and westerly sectors (NW and NO) show distributions with peaks shifted to more negative $\kappa$ values and have longer tails reaching out to larger negative $\kappa$ values, indicating that these sectors show higher sensitivity of terminus position to submarine melting. We test whether the sector-specific $\kappa$-distributions are significantly different using a two-sample Kolmogorov-Smirnov test (Fig. 6b). At the 5% level, the far north sector (NO) is indeed statistically different to all sectors but the north-west (NW),

while the north-west sector (NW) is statistically different to all sectors but the far north (NO) and north-east (NE).

Considering all glaciers together by plotting 5-year binned terminus position anomaly versus submarine melt anomaly (Fig. 7a), variability in submarine melting explains 23% of change in terminus position over all glaciers with a best fit linear coefficient $\kappa = -0.22$. The relationship is statistically significant ($p < 0.01$) after detrending and accounting for autocorrelation, and remains so if points lying more than 2 standard deviations from the trendline are removed (Fig. 7b).





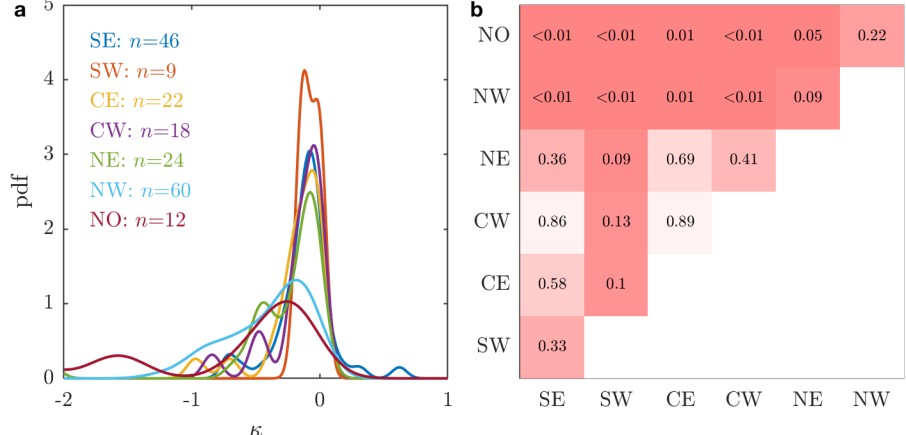

**Figure 6.** Sector-to-sector differences in sensitivity of glaciers to submarine melting. (a) Equivalent of Fig. 5a by ice sheet sector, (b) equivalent of Fig. 5a obtained by removing the indicated sector. In (a) and (b), $n$ indicates the number of glaciers entering the probability density function. (c) $p$-value obtained from a two-sample Kolmogorov-Smirnov test to determine significance of differences between sectors. A value $p < 0.05$ indicates a significant difference between sector distributions in (a), and thus a significant difference in sector sensitivity to submarine melting.

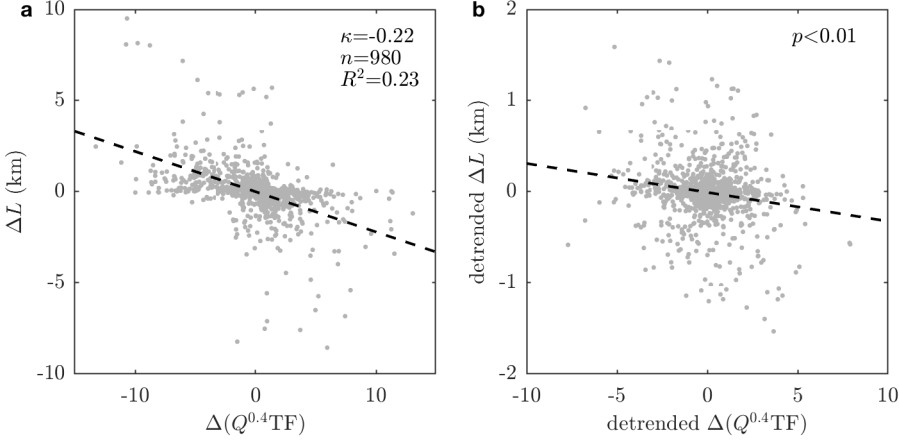

**Figure 7.** Correlation of submarine melt versus terminus position for all glaciers. (a) parameterised submarine melt anomaly versus terminus position anomaly for all glaciers in the dataset. Anomalies are calculated per glacier as the difference from the mean over the full timeseries available for each glacier. Each light grey point is a 5-year binned value. The black dashed line is linear regression on the scatter, and statistics are given in the top right. (b) as for (a), but a linear trend in time for each glacier has been removed from both the submarine melt and terminus position, outliers have been removed, and the $p$-value also accounts for autocorrelation. Since $p < 0.01$, there is a statistically significant relationship between terminus position and submarine melt rate.





Our analysis therefore shows that there is a statistically significant relationship between submarine melting and terminus position at the ice sheet scale, and at a minority of individual glaciers. Similarly, the proposed retreat parameterisation is able to explain a substantial portion of observed terminus position change at the ice sheet scale and at a majority of individual glaciers. Together with process understanding linking submarine melting to glacier dynamics, this analysis supports our proposed pa-

rameterisation. We note however the substantial proportion of terminus position change which is not explained by submarine melting (77% at the ice sheet scale, Fig. 7a), and the wide range of observed glacier sensitivity to submarine melting at both and individual and regional level (Figs. 5a & 6a). As such, we expect that the proposed retreat parameterisation should be able to predict the magnitude of retreat of a population of glaciers in response to climate change, but may perform poorly at an individual glacier level.

## 3.2    Validation of the retreat parameterisation

Given the observed sensitivity of tidewater glaciers to submarine melting (Fig. 5a), there are a couple of ways in which the retreat parameterisation Eq. (1) could be employed. One method would be to project retreat for each glacier using the specific value of $\kappa$ obtained from the history of that glacier. Thus for Kangiata Nunata Sermia we would use $\kappa = -0.11$ (Fig. 4e) and for Store Glacier we would use $\kappa = -0.01$ (Fig. 4j). Under this approach we would be conditioning each glacier to behave in a

similar fashion as it has in the past, so that Kangiata Nunata Sermia would retreat significantly under an increase in submarine melt while Store Glacier would retreat only slightly. This approach might be considered desirable in some circumstances for some time period; for example it is thought unlikely that Store Glacier will retreat significantly in the next few decades (Morlighem et al., 2016).

In general, however, we do not think this is the best approach for centennial timescale projections due to the individuality and

intermittency of glacier response to climate. A certain glacier might appear highly sensitive to melting (with a corresponding high value of $|\kappa|$) because it has retreated through an overdeepening during the period of observation. It might now have stabilised on a bed rock ridge, so that retreat over the next few decades will be much slower than in the recent past, and the high value of $|\kappa|$ would therefore overpredict retreat. Equally, Store glacier may at some point over the next century begin rapid retreat, but if we tune the parameterisation to its past behavior this will not be projected. Similarly, one could consider

employing a sector-specific value for $\kappa$ (Fig. 6a), but this suffers from similar limitations. The value of $\kappa$ becomes more influenced by individual glaciers as the training dataset shrinks (SW has only 9 glaciers and NO has only 12), and thus the future projections become heavily influenced by how individual glaciers have behaved in the past.

Here, we view the distribution of $\kappa$ as a property of the population of tidewater glaciers in Greenland, encompassing the diversity of glacier behavior and response to climate over the recent past and over the full ice sheet. If we were to apply the

retreat parameterisation Eq. (1) with $\kappa$ sampled from the distribution shown in Fig. 5a, then we are conditioning a glacier to behave in the future as the population of glaciers has in the past, rather than as that individual glacier has in the past. In this way we capture the possibility that a glacier will be rather insensitive to climate forcing (because some other glaciers in Greenland experiencing similar climate forcing in the past have retreated only slightly) and the possibility that it will be very sensitive (because some other glaciers have retreated dramatically under similar forcing).



Under this sampling approach, we cross-validate the parameterisation by separating the dataset into training data, on which the parameterisation is calibrated as in Fig. 5a, and test data, on which we test the calibration. This allows the parameterisation to be evaluated by seeing how well it captures observations of retreat when these observations have not been used to calibrate the parameterisation. We consider retreat between 1995-2005 and 2005-2015 as this time period is well-covered by observations

(Fig. 2c). We calculate the mean terminus position in observations for the two time periods and take the difference to give an observed retreat. We project retreat by sampling $\kappa$ from its distribution and, according to Eq. (1), multiplying by the difference in submarine melting between the two time periods. Since Eq. (1) is linear in $\kappa$, the distribution of projected retreat is the distribution for $\kappa$, scaled by the submarine melting anomaly.

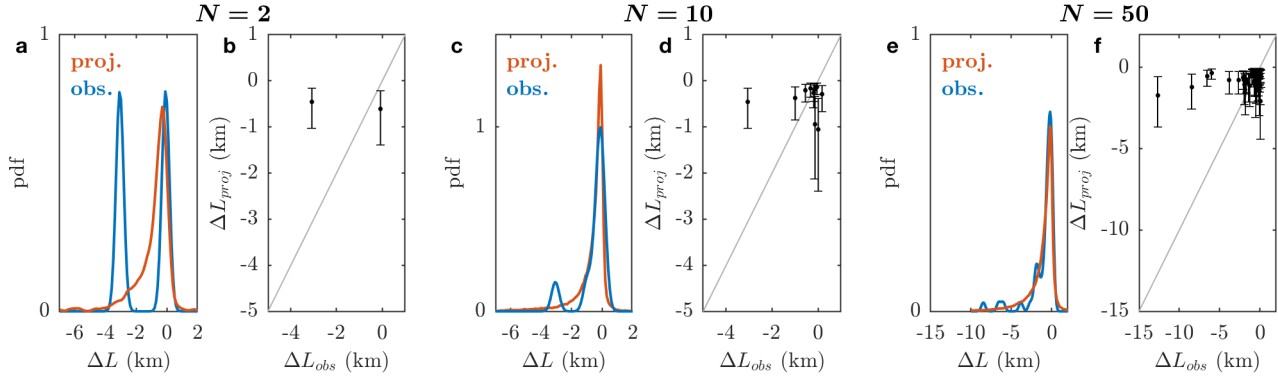

**Figure 8.** Validation of retreat parameterisation on the observed terminus position change between 1995-2005 and 2005-2015. Note that $\Delta L < 0$ indicates retreat. In (a) and (b) we select two glaciers at random to be the test dataset and calculate the distribution for $\kappa$ (as in Fig. 5a) based on all of the remaining glaciers. As described in the text we use the distribution for $\kappa$ to generate projected retreat. In (a) we compare the distributions of observed (blue) and projected (red) retreat. In (b) we plot observed versus projected retreat, where the projection uses the median and interquartile range of the $\kappa$ distribution. (c) and (d) are the same but select 10 glaciers at random, while (e) and (f) select 50 glaciers at random.

The result of this procedure for varying choices of training and test data is shown in Fig. 8. We consider first a test dataset of

2 randomly chosen glaciers, leaving 189 glaciers in the training dataset. The distributions of observed and projected retreat are shown in Fig. 8a (we obtain the observed distribution as a kernel distribution with bandwidth 0.25 km). It is clear that the two distributions do not agree well, a fact which is further illustrated in Fig. 8b – the retreat parameterisation significantly underestimates retreat for one of the glaciers and slightly overestimates retreat for the other. Increasing the size of the test dataset to 10 randomly chosen glaciers (leaving 181 in the training dataset) results in an improved agreement between observations and

projections, illustrated by increased overlap in the observed and projected distributions (Figs. 8c & 8d). Once the size of the test dataset is increased to 50, agreement between the observed and projected distributions is very good. There remain individual glaciers for which the parameterisation performs poorly (Fig. 8f), but the distributions are in very good agreement (Fig. 8e).

These exercises validate the application of the retreat parameterisation when $\kappa$ is sampled from its distribution. They show that given a sufficiently large dataset on which to calibrate the parameterisation, we are able to successfully predict the retreat



of a population of glaciers (Figs. 8e & 8f). Although this sampling approach results in a large range of projected retreat for each individual glacier, it is more honest to the diversity of glacier response than using a single value of $\kappa$ for each glacier. Having justified, calibrated and validated the retreat parameterisation, we now proceed to project retreat over the 21st century.

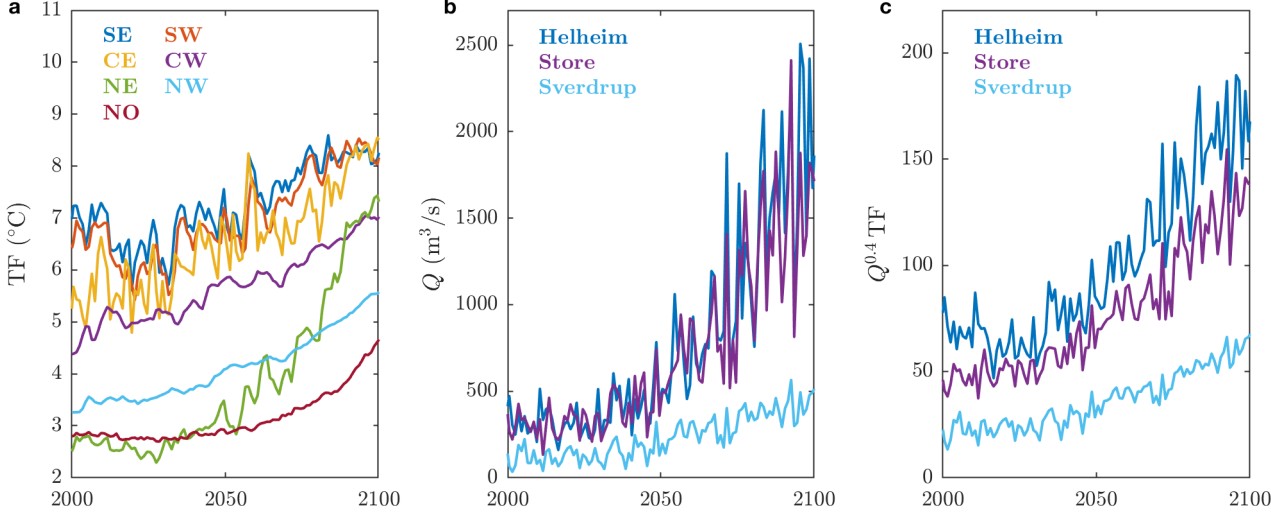

**Figure 9.** Projected RCP8.5 tidewater glacier climate forcing using the climate model MIROC5 and regional climate model MAR. (a) Thermal forcing TF from MIROC5 for each of the ice-ocean sectors. (b) Subglacial runoff $Q$ for selected glaciers from a MAR simulation forced by MIROC5. (c) Parameterised submarine melting. The colors in (b) and (c) show the ice-ocean region to which the glacier belongs.

### 3.3 Projected 21st century tidewater glacier retreat

To demonstrate the use of the parameterisation we consider projected tidewater glacier terminus position change over the 21st century under RCP2.6 and RCP8.5 scenarios in the climate model MIROC5 (Watanabe et al., 2010). To illustrate the procedure, and to highlight spatial variability, we consider three example glaciers (Fig. 9), but once more emphasise the parameterisation is more suited to groups of glaciers, which are considered last of all.

All ice-ocean sectors show significant ocean warming in the MIROC5 RCP8.5 simulation (Fig. 9a), though there is spatial
variability with the far north (NO) showing the least warming by the end of the century ($1.7\,^\circ$C) and the north east (NE) showing the most warming ($3.9\,^\circ$C), more than doubling the thermal forcing in this sector. Three example glaciers also show significant increases in runoff by the end of the century (Fig. 9b). Runoff at Helheim glacier in SE Greenland, averaging $\sim$300 m$^3$s$^{-1}$ during 1995-2014, increases to $\sim$1750 m$^3$s$^{-1}$ during 2081-2100, an increase of approximately a factor of 6. Making the same comparison for Store and Sverdrup glaciers, runoff increases by a factor of 5 and 3.5 respectively. Future
submarine melt forcing, estimated as $Q^{0.4}$ TF, increases by a factor of 2 to 3 by the end of the century in an RCP8.5 scenario (Fig. 9c). In contrast, the climate forcing experienced by tidewater glaciers is projected to change comparatively little under a low greenhouse gas emissions scenario RCP2.6 (not shown).




Tidewater glacier retreat is then estimated by combining the submarine melt forcing for each glacier (e.g. Fig. 9c) with the distribution of glacier sensitivity $\kappa$ (Fig. 5a). We choose to reference our retreat to the year 2014; thus we set $L = 0$ in 2014 for all glaciers, and any projected change $\Delta L$ is to be understood as relative to this baseline. As motivated in section 3.2, the linear coefficient $\kappa$ is sampled $10^4$ times from its distribution (Fig. 5a), creating a distribution of projected retreat for each glacier

(Fig. 10). We do not expect tidewater glaciers would respond sufficiently rapidly to capture the high interannual variability in the climate forcing, and we therefore smooth the projections using a centered 20-year moving average. Although this is a longer smoothing interval than the 5-year binning applied to the calibration datasets, it targets the century-scale trend which is the focus of this paper and current ice sheet modeling efforts.

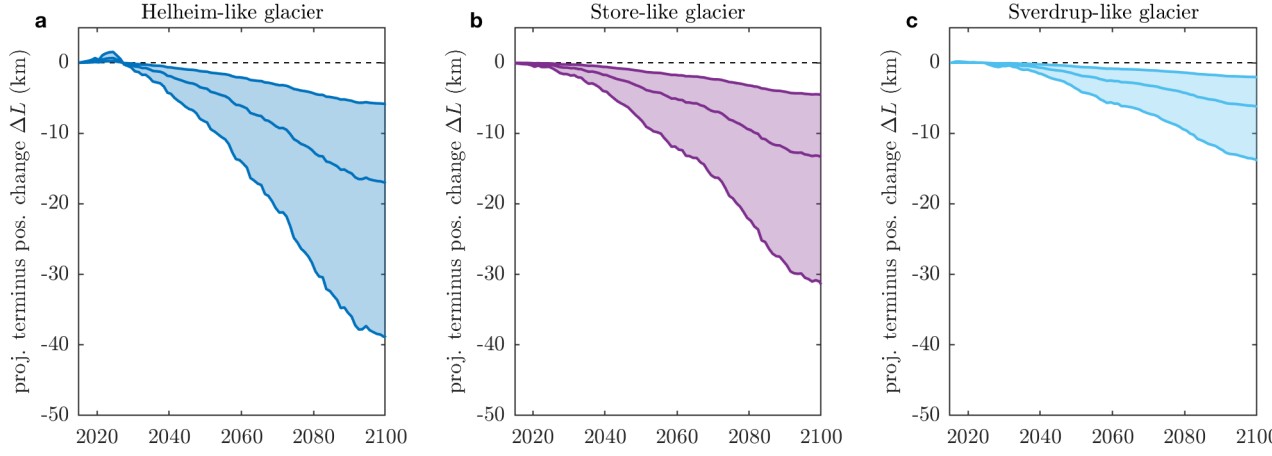

**Figure 10.** Projected terminus position change $\Delta L$ under an RCP8.5 scenario with climate forcing from MIROC5 as shown in Fig. 9. Three example glaciers are shown, but we emphasise that these projections suggest what would happen to a typical glacier experiencing the particular climate forcing of these glaciers, since glacier-specific factors will strongly modulate retreat. As described in the text, the projections have been smoothed using a centered 20-year moving average. The three projections on each plot show the 25th, 50th and 75th percentile retreat. In sum, we expect that a glacier experiencing the climate forcing of Helheim glacier has a 50% chance of experiencing retreat which falls into the shaded region in (a).

Given the important role played by glacier-specific features such as bed topography at individual glaciers, it is most appro-

10 priate to think of the projections in Fig. 10 as projections for an average glacier experiencing that particular climate forcing. Or more precisely, if there were many glaciers experiencing the same forcing as Helheim (Fig. 10c), we would expect retreat for 50% of those glaciers to fall in the shaded region in Fig. 10a. With this caveat in mind, Helheim glacier has the largest projected retreat of the three example glaciers (Fig. 10a) because it experiences the largest increase in submarine melt (Fig. 9c), due to a large increase in subglacial runoff (Fig. 9b) and a smaller increase in thermal forcing (Fig. 9a). The median retreat for

Helheim by 2100 is ∼17 km, with an interquartile range of 6-39 km. Significant retreat is not projected until after 2030 because the period 2014-2030 is relatively cold in the MIROC5 RCP8.5 simulation (Fig. 9), likely due to multidecadal variability in





the climate model. Sverdrup glacier shows the smallest projected retreat (Fig. 10c) because the absolute increase in melting is smallest (Fig. 9c); the median retreat by 2100 is ∼6 km while the interquartile range is 2-14 km.

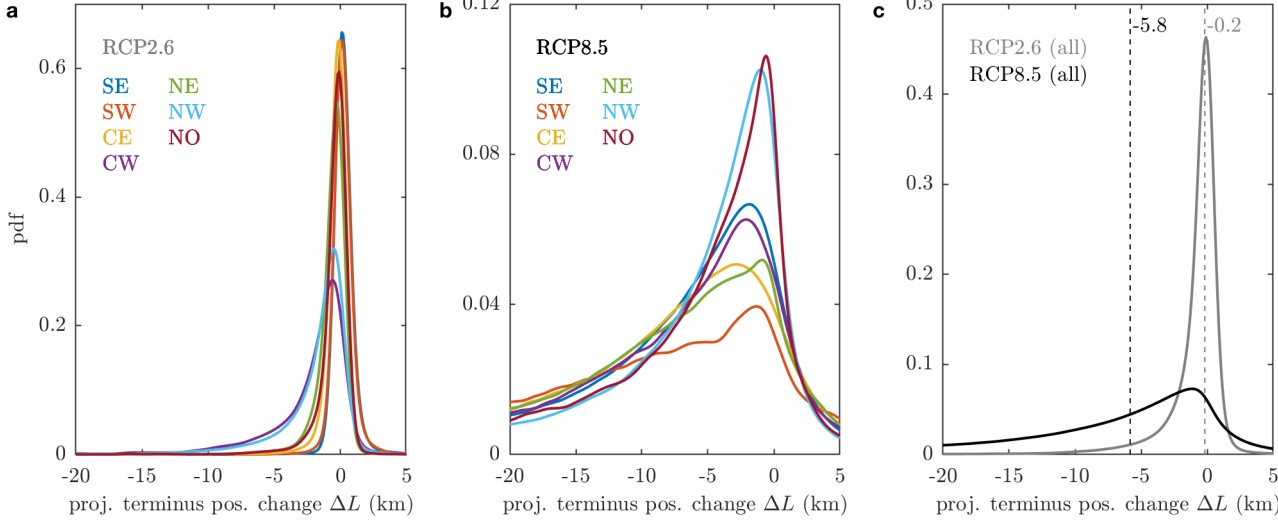

**Figure 11.** Probability distributions for Greenland tidewater glacier terminus position change by 2100. (a) shows an RCP2.6 scenario by sector, (b) shows an RCP8.5 scenario by sector while (c) shows full ice sheet distributions for both RCP2.6 and RCP8.5. The vertical dashed lines and number labels on (c) indicate the median retreat in km for each emissions scenario. Note the differing vertical axes on each plot.

We then repeat this procedure for each of the 191 glaciers in our dataset and group the glaciers by ice-ocean sector to give probability distributions for projected terminus position change by 2100 for each sector under both RCP2.6 and RCP8.5

scenarios (Fig. 11). Sector-to-sector variability in the projections arises both due to regional variability in projected climate and the characteristics of glaciers in a sector, as larger glaciers tend to retreat further under our parameterisation. Under an RCP2.6 scenario, projected change is tightly clustered around 0 because MIROC5 predicts little warming of the ocean or atmosphere in this scenario (Fig. 11a). For most regions, glacier advance ($\Delta L > 0$) is as probable as retreat, while in the CW and NW sectors retreat is more probable because MIROC5 does predict significant warming of the ocean in these regions (not shown). Under

an RCP8.5 scenario, sector retreat distributions become much broader with much larger retreat much more probable (Fig. 11b). The distribution for the NW sector is fairly sharply peaked because this region has a large number of similar-sized glaciers, and in contrast the SW sector shows the broadest distribution because it has a small number of glaciers of diverse size.

At the ice sheet scale, projected terminus position change under RCP2.6 is sharply peaked around 0 with a median retreat of 0.2 km, indicating little change from present with a significant number of glaciers predicted to advance (Fig. 11c). Thus

under RCP2.6 we expect that 50% of glaciers in Greenland would undergo retreat of less than 0.2 km by the end of the century. There is a not insignificant tail of retreat greater than 3 km which is almost entirely due to significant ocean warming in the CW and NW sectors (Fig. 11a), though this may be specific to the climate model MIROC5 (Yin et al., 2011). Under RCP8.5 the distribution shifts leftwards and broadens significantly (Fig. 11c). The peak of the distribution still occurs at a moderate





retreat of only ~1 km, because there are a large number of small tidewater glaciers in Greenland which typically show only small response to climate forcing. The median retreat is now 5.8 km; thus we are suggesting that 50% of glaciers will undergo retreat exceeding 5.8 km by 2100. The distribution also suggests that retreat by 2100 will exceed 10 km for 35% of glaciers, and will exceed 20 km for 19% of tidewater glaciers in Greenland under an RCP8.5 scenario.

## 4   Discussion

### 4.1   Philosophy and interpretation of parameterisation

We have used past climate and terminus position observations and reconstructions, together with process understanding of submarine melting and tidewater glacier dynamics, to show there is a statistically significant relationship between tidewater glacier terminus position and estimated submarine melt. On this basis we have calibrated and validated a parameterisation in which tidewater glacier retreat is linearly related to submarine melting. The parameterisation is not intended to capture short-term glacier-to-glacier variability in retreat rate, which is likely driven by bed topography or fjord dynamics. As such the parameterisation is essentially asking, given past climate and terminus position variability, and projected climate warming, how much should we expect tidewater glaciers to retreat? Our strategy emphasises distributions of retreat rather than individual glacier retreat trajectories.

We consider the existence of a statistically significant correlation between terminus position and parameterised submarine melting to strengthen the argument for the importance of submarine melting, but it is not inconceivable that the retreat parameterisation might be inadvertently accounting for other drivers of calving. For example, the structural integrity of ice mélange and sea ice, thought to be important in some locations for inhibiting calving (Amundson et al., 2010; Christoffersen et al., 2012; Moon et al., 2015), would likely be compromised by increased air temperature (and thus increased runoff $Q$) and increased ocean temperature (and thus increased ocean thermal forcing TF). Furthermore, because the ocean and atmosphere are a coupled system, the time series of runoff $Q$ and ocean thermal forcing TF are not necessarily independent, which further confounds the identification of the key processes driving ice sheet change.

One implication of the form we have assumed for the retreat parameterisation is that the terminus position is always in equilibrium with the climate, i.e. if the climate stabilises then the terminus position stabilises, and there is no continuing or lagged impact from past climate. The timescale of response of tidewater glaciers to climate is an ongoing topic of research, and through binning terminus positions and climate data in 5-year intervals, and by smoothing projections with a 20-year moving average, we have here implicitly assumed a terminus position response timescale of 5-20 years, which is supported by the observed rapid changes at tidewater glaciers in recent decades (Straneo and Heimbach, 2013) and by theory (Robel et al., 2018). Stabilisation of the terminus position does not however imply stabilisation of mass loss, because thinning can propagate up-glacier for decades after terminus retreat (Price et al., 2011) and this would be captured by an ice sheet model employing our retreat parameterisation. But a slower lagged response arising for example from atmospheric-driven thinning propagating down-glacier to the terminus and leading to frontal retreat (Robel et al., 2018) would not be captured by our parameterisation.




## 4.2 Use of parameterisation

We envisage this retreat parameterisation and projections of retreat to be of use primarily to ice sheet modelers looking to simulate ice sheet response to outlet glacier retreat. The principal advantages of the parameterisation are its simplicity and its empirical validation, thus the critical interaction of the ice sheet with the ocean can be represented in a manner which is

consistent with observations and scales well to practical applications. Such an approach is clearly very different in character from simulations which explicitly try to resolve glacier frontal dynamics (e.g. Nick et al., 2013; Morlighem et al., 2016); indeed terminus positions in such studies typically jump quickly from one stable position to the next, while our projections instead give gradual retreat. The rapid transition between stable positions is evident in observations (e.g. Catania et al., 2018), and certainly accurately projecting mass loss at individual glaciers and over short timescales means accurately modeling these transitions.

We posit however that imposing gradual retreat, as suggested here (e.g. Fig. 10) is a reasonable approach for capturing mass loss at an ice sheet scale and over multidecadal timescales, especially since the timing of rapid terminus transitions is hard to capture.

Use of a retreat parameterisation does heavily parameterise tidewater glacier frontal processes, but we emphasise that it does not place any constraints on ice thickness or velocity at the ice-ocean boundary, which would still be calculated dynamically

by the ice flow model. The total dynamic sea level contribution is then the sum of the ice above flotation removed by the retreat parameterisation together with the inland propagation of thinning in response to retreat. The retreat parameterisation described in this paper is therefore the standard approach that has been recommended to ice sheet modelers simulating the future of the Greenland ice sheet in the ISMIP6 project (Nowicki et al., 2016), the results of which will feed into sea level projections in the next IPCC assessment report. Specifics relating to ISMIP6 are detailed in Appendix B.

## 4.3 Comparison to existing projections

Few studies have projected tidewater glacier retreat for comparison to our projections. Nick et al. (2013) used a flowline model to project retreat for four of Greenland's largest glaciers under an RCP8.5 scenario. We compare projections in Fig. S5; for all glaciers the projections in Nick et al. (2013) lie within the interquartile range of our projections (though we do not consider Peterman glacier here as it has a persistent ice shelf). Taking Helheim glacier as an example, Nick et al. (2013) project retreat

of 17-26 km between 2014 and 2100 while here we project a median of 17 km and an interquartile range of 6-37 km (Fig. 10). Beckmann et al. (2018) used a similar flowline model to project retreat for 12 assorted glaciers under an RCP8.5 scenario; a comparison is shown in Fig. S6 and shows once more that - within the interquartile range - our projections agree with all 12 of those from the flowline model. Lastly, Morlighem et al. (2019) used a state of the art ice flow simulation together with dynamic modeling of frontal processes to project the future evolution of NW Greenland, but their climate forcing is not easily

comparable to the RCP scenarios considered here.

Cowton et al. (2018) calibrated a retreat parameterisation based on 20 years of evolution of 10 tidewater glaciers in east Greenland. Relative to their study, we have substantially greater spatial coverage (including 191 of the largest tidewater glaciers in Greenland) and significantly greater temporal coverage (including 126 glaciers with a record longer than 20 years). This





expansion of the dataset allows us to find a statistically significant link between submarine melting and terminus position at the ice sheet scale, to generate projections for every tidewater glacier and region in Greenland, and to quantify uncertainties in forward projections by sampling from a large distribution of glacier sensitivity to submarine melting. Nevertheless, if we compare RCP2.6 and RCP8.5 projections using the parameterisation from this study and from Cowton et al. (2018), we find

they are very similar (Fig. S7). The parameterisation in Cowton et al. (2018) predicts a median retreat of 0.15 km under RCP2.6, relative to 0.2 km here (Fig. 11c), and a median retreat of 6.6 km under RCP8.5, relative to 5.8 km here (Fig. 11c).

### 4.4  Possible improvements

We have assumed a linear relationship between submarine melting and terminus position. While there is significant evidence linking tidewater glacier retreat to increased submarine melting (e.g. Straneo and Heimbach, 2013, and references therein),

process studies do not yet indicate a simple relationship between submarine melting and calving, or submarine melting and terminus position (Benn et al., 2017b). Luckman et al. (2015) and How et al. (2019) suggest that at two glaciers in Svalbard, calving is largely restricted to failure of ice which is undercut by submarine melting, so that frontal ablation is linearly paced by submarine melting. This simple relationship may however be limited to glaciers where submarine melting can outpace ice flow, which is not thought to be the case for most of Greenland's larger tidewater glaciers (Carroll et al., 2016). At faster-flowing

glaciers, studies have been conflicted on the importance of submarine melting (Cook et al., 2014; Krug et al., 2015; Todd et al., 2018) while other studies show a highly non-linear response of calving to submarine melting (Benn et al., 2017a; Ma and Bassis, 2019). Given this uncertainty, we have here assumed the simplest possible linear relationship, and indeed we find that this is statistically significant (Fig. 7b). We do not however rule out the possibility that non-linear relationships, or different combinations of climate forcing, possibly including additional parameters such as grounding line depth, might provide a closer

relationship between forcing and retreat, ultimately feeding through to reduced uncertainty in future projections.

One could also consider a retreat parameterisation based on relative, rather than absolute, change in submarine melting. It may be that the apparent increased sensitivity of glaciers in NO and NW Greenland to submarine melting (Fig. 6) results from initially low submarine melt rates in those regions, such that a given absolute increase in submarine melting is a larger relative increase in NO and NW Greenland than further south. We do however suspect that the formulation in terms of absolute melt rate

may be key to finding a statistically significant link between submarine melting and terminus position. The absolute formulation encapsulates the fact that in general, larger glaciers have greater potential to undergo large retreat. Equivalently, the numerous small tidewater glaciers in Greenland, which flow at speeds of a few hundred metres per year (Rignot and Mouginot, 2012), are unlikely to undergo retreat of several kilometers on short timescales. This is captured by an absolute formulation because small glaciers have small hydrological catchments, small runoff $Q$, small submarine melt rates and therefore limited absolute

variability in submarine melt rate and projected retreat. In this sense, the runoff $Q$ appearing in the retreat parameterisation could be thought of as a 'glacier size' parameter, and we speculate that this consideration of glacier size may be critical to finding significant relationships between glacier behaviour and climate, which may explain why some studies have found significant relationships (Cowton et al., 2018; Porter et al., 2018; Cook et al., 2019) and others have struggled (Murray et al., 2015; Carr et al., 2017).





Another possibility would be to formulate a parameterisation for frontal ablation rate rather than retreat (the two being related by retreat rate = ice velocity – frontal ablation rate). A parameterisation for frontal ablation rate might be considered preferable because then, through an ice sheet model, the ice velocity is allowed to influence the terminus position - that is, one could capture the potential feedback where retreat is stabilised due to an increase in ice velocity. We have nevertheless chosen to form a retreat parameterisation for three key reasons. First, there is the pragmatic fact that a longer time series of terminus positions is available than of frontal ablation rate. The latter requires ice velocity, which is in general hard to obtain before 2000. Second, because we are tuning our parameterisation empirically, the parameterisation in some sense includes all potential feedbacks, because these feedbacks will have been influencing the terminus positions which enter our calibration. Third, we note that Haubner et al. (2018) showed at Upernavik Isstrom that by imposing externally-specified terminus positions, it is possible to capture dynamic mass loss of the glacier behind, suggesting that specifying terminus position through a retreat parameterisation is a feasible approach to modeling ice sheet dynamic response to climate.

We assess there to be three key areas in which the current parameterisation could be improved. Firstly, the huge importance of a handful of large glaciers to Greenland's sea level contribution (Enderlin et al., 2014) motivates the need for a retreat parameterisation which performs well at an individual glacier level rather than just over a population of glaciers. This requires consideration of bed topography and we therefore place high priority on exploring simple ways of including bed topographic effects in a similar retreat parameterisation to that considered in this study, stabilizing glaciers on pinning points and promoting rapid retreat through overdeepenings. Secondly, more long-term observations would be hugely valuable for improving the calibration and validation of the parameterisation. Aerial photography (Bjork et al., 2012) is a promising source of long-term terminus position records, but long-term oceanographic observations are sparse; careful use of limited historical records or reanalysis products might prove fruitful. Thirdly, while we expect that our estimates of subglacial runoff $Q$ entering the parameterisation are accurate (e.g. Langen et al., 2015; Noël et al., 2018), the thermal forcing TF is highly simplified and thus less certain, being based on spatial and depth averaging of ocean temperatures over the continental shelf and beyond. Through fjord dynamics and fjord-shelf exchange, thermal forcing at the calving front may differ from that on the continental shelf (e.g. Gladish et al., 2015a), and may differ at adjacent glaciers (Bartholomaus et al., 2016). There is therefore an urgent need for methods that can translate offshore ocean properties to calving front thermal forcing, and a pressing need for sustained oceanographic observations with which to validate these models.

## 5 Conclusion

We have used surface melt output from a regional climate model, compilations of ocean temperature, and records of glacier retreat to examine links between parameterised submarine melting and tidewater glacier terminus position change since 1960 for 191 of Greenland's marine-terminating glaciers. We find a statistically significant relationship between parameterised submarine melt rate and terminus position at the ice sheet scale, and that variability in submarine melting can explain more than 50% of variability in terminus position at 105 of the 191 glaciers considered.





On this basis, we develop a simple parameterisation relating tidewater glacier retreat to submarine melt anomaly, providing a method of capturing the critical interaction between the ice sheet and ocean, and the dynamic response of the Greenland ice sheet to tidewater glacier retreat, without the huge computational expense of explicitly resolving calving processes. The parameterisation is weakest when applied to an individual glacier over short timescales, when glacier-specific factors such as

bed topography play a dominant role in determining whether, when, and how much a glacier will retreat in response to a climate forcing. The parameterisation is strongest when applied to a population of glaciers, for example an ice sheet region when it provides an envelope of projected retreat given how sensitive tidewater glaciers have collectively been to climate forcing in the recent past.

We provide example projections under low (RCP2.6) and high (RCP8.5) greenhouse gas emissions scenarios using output

from a single global climate model MIROC5. Since significant variability exists between climate models (e.g. Yin et al., 2011), these projections should be considered largely as an illustration. For the low emissions scenario, tidewater glaciers show, in general, little change by the end of the century. Under the high emissions scenario, ocean thermal forcing increases by 2-4 °C and subglacial runoff increases by a factor of 3-6 by 2100. In response, we project a median Greenland tidewater glacier retreat of 5.8 km, and suggest that 35% of glaciers will retreat more than 10 km, and 19% will retreat more than 20 km by the end of

the century.

The analysis and parameterisation described in this study forms the standard method that has been recommended to ice sheet modelers taking part in the ISMIP6 project (Nowicki et al., 2016), which aims to project sea level contribution from Greenland for the IPCC AR6. We believe that this simple process-motivated parameterisation will prove useful for the projection of dynamic mass loss from Greenland, and expect that it will be complemented by more complex approaches as our understanding

and modeling and tidewater glacier dynamics continues to improve.

*Data availability.* Terminus positions may be downloaded from https://nsidc.org/data/nsidc-0642/ (last access October 2018). All other terminus position and ice flux datasets may be requested from the sources summarised in Table S1 and Appendix B. Information on the RACMO2.3p2 SMB data can be found at http://www.projects.science.uu.nl/iceclimate/models/greenland.php (last access April 2019). EN4.2.1 oceanographic data is available at https://www.metoffice.gov.uk/hadobs/en4/download.html (last access April 2019). MIROC5

model output is available at https://esgf-node.llnl.gov/projects/esgf-llnl/ (last access April 2019). The MAR based future runoff projections are available on ftp://ftp.climato.be/fettweis/MARv3.9/ISMIP6/GrIS/ (last access April 2019). Further information on the ISMIP6 project may be found at http://www.climate-cryosphere.org/activities/targeted/ismip6 (last access April 2019).

## Appendix A: Climate model bias correction

We ensure that the runoff and thermal forcing coming from the climate model MIROC5 and MAR forced by MIROC5 are

roughly correct in the present day by bias-correcting both time series. Specifically, we subtract a constant offset from both time series which is given by comparing the time series to 'observations' over the period 1995-2014. Thus the final forcing time





series are defined as

$$X(t) = X_{MIROC5}(t) - [X_{MIROC5}(1995 - 2014) - X_{OBS}(1995 - 2014)]$$

where $X$ is either runoff $Q$ or thermal forcing TF, 'OBS' refers to either RACMO or EN4, and '1995-2014' means taking an average over this time period. As a result, the average of $X(t)$ over the period 1995-2014 will agree with the 'observations' from

RACMO or EN4 over the same time period. Biases are calculated per glacier for runoff and per sector for ocean temperature. Typical runoff biases are $\sim$20 m$^3$ s$^{-1}$, as compared to interannual variability of $\sim$60 m$^3$ s$^{-1}$ in the MAR forced by MIROC5 projections. Thus the normalised bias, defined as the bias divided by the interannual variability, is typically less than 0.5 and therefore considered small. In contrast, typical ocean temperature biases are $\sim$0.5-1 °C compared to interannual variability of $\sim$0.2 °C in MIROC5, so that the ocean temperature bias corrections are significant.

**Appendix B:  ISMIP6 considerations**

We consider how these retreat projections can best be implemented in the ISMIP6 process (Nowicki et al., 2016). ISMIP6 is an international collaborative effort to run a suite of ice sheet models forced by a suite of climate models to estimate sea level contribution from the Greenland and Antarctic ice sheets for the coming Intergovernmental Panel on Climate Change Assessment Report (IPCC AR6). Further information may be found at http://www.climate-cryosphere.org/activities/targeted/ismip6. This

study is relevant only for the ocean forcing aspect of Greenland ice sheet models. The ice sheet models taking part are diverse in terms of the processes they can represent, the way they are initialised, and their spatial resolution (Goelzer et al., 2018). One of the key challenges to meet then is to identify a method of implementing ocean forcing which works for every model in a consistent manner. The simplicity of the retreat parameterisation described in this study makes it ideal for this purpose, and it is thus the standard method recommended to ice sheet modelers taking part in the exercise.

The retreat parameterisation essentially provides time series of retreat that can be imposed on an ice sheet model. One immediate difficulty is that the 'present day' state of the ice sheet can look very different in each ice sheet model taking part due to differing initialisation methods and resolutions; some models have 10% too much ice by volume in their initial state (Goelzer et al., 2018). In order to avoid artificially high sea level contributions due to differences in model initialisation, we are proposing that the parameterised retreat be imposed relative to the initial extent of the ice sheet, rather than defining absolute

ice sheet extent. We also need to define over what spatial scale the retreat parameterisation is defined – imposing glacier-specific retreat at individual glaciers might be desirable, but because the initial ice sheet extent is not the same in model and observation, it is not guaranteed that a model will have the same glaciers as reality. In particular, a model which has an initial ice extent which is too large, or a model which has low resolution, might have merged two glaciers which are distinct in reality. Imposing retreat per glacier would require accounting for these situations dynamically as the ice sheet simulations progress. To

remedy this, we suggest imposing the retreat parameterisation resolving only each ice sheet-ocean sector (Fig. 3) – this avoids difficulties associated with differing ice sheet extent and is also more consistent with the focus on populations of glaciers rather than individual glaciers in this study.





An immediate issue however, is that Greenland's sea level contribution, which is the ultimate goal of ISMIP6, may be dominated by a small number of large glaciers. These glaciers are likely to experience retreat which is larger than the median retreat per sector (Figs. 10 & 11). Thus applying the median retreat per sector might systematically underestimate retreat for these large glaciers which matter the most for ice sheet mass change. For ISMIP6, we therefore propose to apply retreat which

is an ice flux-weighted mean of the projected retreat for all glaciers in a sector. We therefore use the mean observed 2000-2010 ice flux (Enderlin et al., 2014; King et al., 2018) to provide weightings for the retreat projections for each glacier, so that the largest glaciers in each region are ascribed high relative importance. Specifically, we project retreat $r_j(t)$ for each individual glacier $j$ by sampling from the $\kappa$ distribution and combining $\kappa$ with the submarine melt anomaly. We project the flux-weighted mean retreat $R_i(t)$ for each ice sheet-ocean sector $i$ as

$$R_i(t) = \sum_j f_j r_j(t) / \sum_j f_j$$

where $f_j$ is the observed mean 2000-2010 ice flux of glacier $j$, and the sum runs over every glacier $j$ in the sector $i$. We repeat this procedure $10^4$ times to generate an ensemble of projected retreats for each sector. We smooth the time series (as in Fig. 10) using a 20-year centered moving average, finally identify the 25th, 50th and 75th percentiles of the ensemble, and call these the 'low', 'medium' and 'high' projected retreat for each sector.

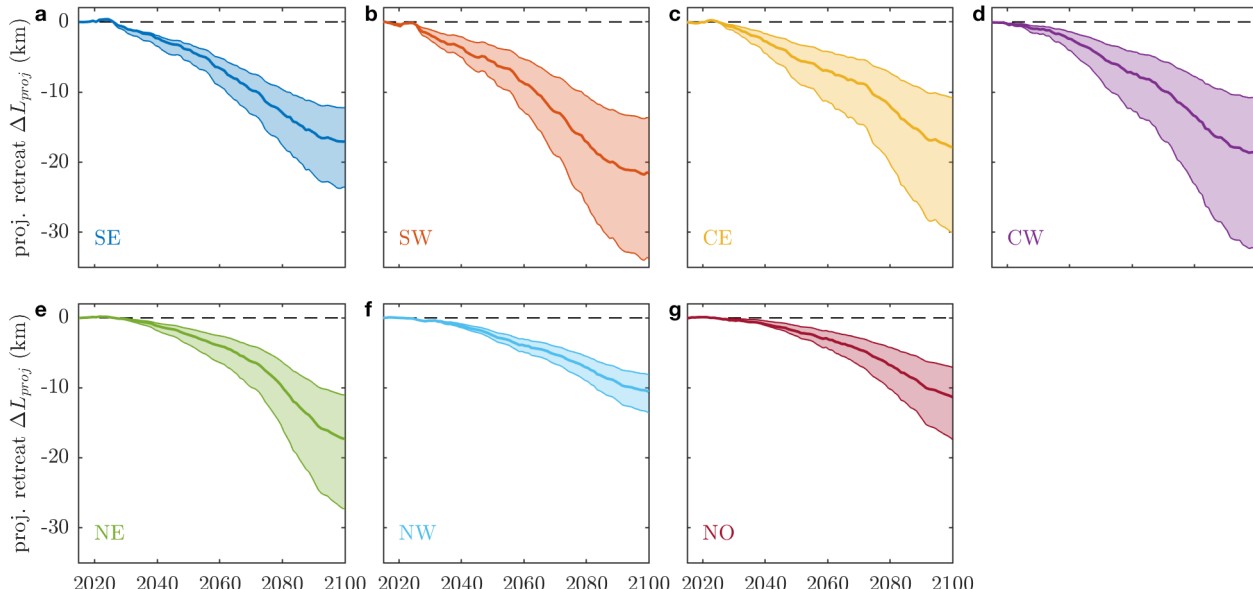

**Figure B1.** Proposed ISMIP6 terminus retreat by ice-ocean sector under an RCP8.5 scenario as projected in the climate model MIROC5. As described in the text, the trajectories are formed from the ice flux-weighted mean of individual glacier projections. The solid lines show the 25th (high), 50th (medium) and 75th percentile (low) retreats in each sector.

The resulting projections for the climate model MIROC5 under emissions scenario RCP8.5 are shown in Fig. B1. Relative to Fig. 11 (which was simply the distribution for retreat in each region without taking any averages or flux-weighting), taking




flux-weighted means increases the projected retreat in all regions. As already described, this arises because we are ascribing greater weight to the largest glaciers, which, under the retreat parameterisation, tend to retreat further. The interquartile range in Fig. B1 generally shrinks relative to Fig. 11 because we are taking a mean. This is especially true for NW Greenland, which has a large number of glaciers with a similar ice flux, but is less so for CW Greenland, because the projected retreat becomes

very heavily influenced by Jakobshavn Isbrae alone.

This ice-flux weighting ensures that the retreat projections are optimised for the estimation of sea level contribution from Greenland, and it is these retreat timeseries that we suggest are used in ISMIP6. It is also proposed that models run each of the 'low', 'medium' and 'high' scenarios to sample uncertainty arising from the retreat parameterisation approach. Finally, we have here presented projections using the climate model MIROC5. This is one model of many in the CMIP5 ensemble, and

it is proposed that additional ice sheet simulations are run in which the retreat parameterisation receives climate forcing from other climate models.

*Author contributions.* DS collated the datasets and undertook the analysis with input from all authors. FS led the ocean working group of the ISMIP6 collaboration. DF provided past and future subglacial runoff over glacier drainage basins. CL provided MIROC5 ocean outputs and expertise on CMIP climate models. HG provided ice sheet modeling expertise to ensure the retreat parameterisation is implementable

and extended ice sheet basin delineations over the ocean. XF ran MAR simulations forced by CMIP models. JH evaluated the EN4 ocean climatology. All authors discussed the results. DS wrote the manuscript with input from all authors.

*Competing interests.* Xavier Fettweis is a member of the editorial board of the journal.

*Acknowledgements.* Donald Slater, Fiamma Straneo and Jamie Holte were supported by NSF grants 1916566 and 1418256, and by NASA grant NNX17AI03G. Denis Felikson acknowledges financial support from the NASA Postdoctoral Program. Chris Little acknowledges

financial support from NSF grant 1513396. Heiko Goelzer has received funding from the programme of the Netherlands Earth System Science Centre (NESSC), financially supported by the Dutch Ministry of Education, Culture and Science (OCW) under grant number 024.002.001. Computational resources for performing MAR future projections have been provided by the Consortium des Équipements de Calcul Intensif (CÉCI), funded by the Fonds de la Recherche Scientifique de Belgique (F.R.S.–FNRS) under grant no. 2.5020.11 and the Tier-1 supercomputer (Zenobe) of the Fédération Wallonie Bruxelles infrastructure funded by the Walloon Region under the grant agreement

no. 1117545. We thank Camilla Andresen, Nadine Steiger, James Lea, Konstanze Haubner, Ginny Catania, Tom Cowton, Charlie Bunce and Rachel Carr for providing terminus position datasets. Thanks to Brice Noël for RACMO2.3p2 output, to Ellyn Enderlin and Michaela King for ice flux datasets, and to Jeremie Mouginot for sharing ice sheet basin delineations. All members of the ISMIP6 collaboration are thanked for discussions and feedback, with particular thanks to Sophie Nowicki, Mathieu Morlighem, Hélène Seroussi, Alice Barthel and Tim Bartholomaus.





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
