# Peer review of "Estimating Greenland tidewater glacier retreat driven by submarine melting"

_The Cryosphere, 2019_

## Referee Comment (RC1) · Anonymous Referee #1 · 24 May 2019

**Review of: Past and future response of Greenland's tidewater glaciers to submarine melting**

**General comments**

This paper presents a new parameterization for Greenland marine-terminating glacier retreat based on subglacial runoff and ocean thermal forcing. The authors use data from 1960-present to develop and test the parameterization, and model projections to 2100 to examine the parameterization projections to 2100. Overall, the paper is well written and easy to follow. Data and methods are explained in appropriate detail with useful and clear figures. Apart from the comments below the paper is thorough in discussing the assumptions, limits, opportunities, and other research related to this contribution. The paper will be a valuable reference for ice sheet modelers and the cryosphere community, particularly given that it outlines the recommended parameterization method for ISMIP6. It also provides an excellent discussion of the methods and opportunities for improvement, and the paper will be an important launching point for further research. Progress on incorporating ice sheet-ocean interaction into ice sheet models is critical, and the step provided in this paper is welcome.

I do have some concerns about discussion of projections made in the paper and think that adjustments are needed in the discussion and messaging to ensure that the authors are not overstepping the limits of the science. Throughout the paper the authors emphasize that there is a statistically significant relationship between terminus change and submarine melt (here  $Q^{0.4}$ TF), but Figure 7b also highlights how tenuous this relationship is, and the authors results themselves also demonstrate this (e.g., how much of terminus change is explained by submarine melt, etc.). It is clear that the authors are well aware of this, as they emphasize multiple times that the parameterization is best for groups of glaciers (or even an ice-sheet-side grouping), and that there are other components of the system, like topography, that may play a similar or larger role in determining retreat. Despite these statements, however, the authors still move forward on including projections for individual glaciers as well as ice-sheet-wide projections regarding retreat. These items are misleading and likely to cause confusion and misuse by readers who are not as familiar with the details of marine-terminating systems or are coming to the paper specifically for this projected information but with little attention to the paper details. By including these details (inappropriately I suggest), the authors are providing projections that can too easily be taken out of context or cited without inclusion of the nuances and caveats that appear in this manuscript. My feeling is that because there is a statistically significant relationship, the authors have in several instances pushed their statements beyond what is actually justified by the data (and especially considering that the parameterization does not account for other major known influences, like topography). There are two areas that are particularly problematic:

 The title. The main point of this paper is a new parameterization. The paper is not providing new and improved details on past glacier response to submarine melting nor future projections with a confidence level that warrants the current title. A more appropriate title would be something along the lines of: 'New parameterization for incorporating Greenland tidewater glacier response to submarine melt in ice sheet models' or 'Parameterizing Greenland tidewater glacier response to submarine melting for improved ice sheet projections'

2) Figure 10 and related discussion of projections. First, labeling on Figure 10 is misleading. 'Helheim-like' (and similar) has little functional meaning – like Helheim in terms of climate, bed topography, shape, ice flux, etc.? While the caption points out that these are projections that are more appropriately commenting on regional glacier behavior, the figure and elements of the discussion (e.g., 16/15 specifically states 'retreat for Helheim...') do not reflect this. Second, given the limits to the technique, I find little justification for discussing projections of individual glaciers – or even of ice-sheet-wide glaciers - in any context. Using this parameterization alone to project future glacier behavior (e.g, 17/15) is problematic.

Also, the broader issue of overstating the conclusions in ways that do not reflect the full limits of the parameterization does rear its head in a few other places, and I have tried to capture those in the specific comments below. Once these elements of the discussion align appropriately with the actual skill (and known limits) of the parameterization, this paper will be an excellent addition to The Cryosphere.

One other general comment – I would like to see some discussion regarding the assumption that surface topography is unchanged. This assumption is made for developing the parameterization from past data, and in projecting future changes. Substantial peripheral thinning across Greenlandis well documented, however, and increasing future runoff suggests that this trend will continue or accelerate. Ice thickness is an important component for retreat, and I am concerned that without considering it in future projections, the authors are omitting an important system change, but it is not as well acknowledged or discussed as some other omitted factors.

**Specific comments (by page/line number)**

4/29. Specify if you are using the mean, weighted mean, etc. over the 5-year bins.

8/18. Label Kangerdlugssuaq in one of the figures.

8/24+. It is a bit confusing here to start by saying runoff and TF are estimated using MIROC5, but then explain that runoff is actually from MAR forced by MIROC5. Better to simplify and streamline here by going straight for – runoff from MAR forced by MIROC5 and TF from MIROC5.

11/14. Changing 'but' to 'except' makes this sentence easier to understand.

15/17. Consider including plots related to the RCP2.6 scenario in the supplementary material. These can be valuable for communicating the substantial differences in retreat under an RCP2.6 v. RCP8.5 scenario.

13/30. In essence, this parameterization assumes that the *setting* of a region of glaciers – and by this I mean all of the other elements influencing behavior that are outside of submarine melt – will remain the same in the future as it has been in the past. This is what provides regional or ice sheet scale confidence for using the parameterization in projections. I think it would be appropriate for the authors to more clearly acknowledge this. Given the importance of

topography for short-term AND long-term retreat behavior, it is important to raise this issue as intrinsically tied to the assumptions made regarding future terminus change.

17/11. Given the parameterization, 'similar-sized glaciers' is more correctly 'similar-sized glacier basins'.

18/11. While bed topography does control short-term variability, it is also unquestionably a major factor in determining long-term change when it comes to patterns of retreat. For example, the multi-centurial behavior a glacier that extends into a long overdeepening (e.g., Jakobshavn) will be dramatically different than a glacier that may retreat onto shallow topography or even become land-terminating.

18/32. Thinning may also occur directly at the terminus from SMB changes.

19/3-5. This sentence is overly confident. Perhaps something like: 'The principal advantages of the parameterization are its simplicity and context provided via empirical validation, thus the critical interaction of the ice sheet with the ocean can be represented in a manner which is informed by observations and scales to large region and ice-sheet-wide applications.'

21/10. Delete 'behind'

21/12 & 17, 22/3. Consider removing 'huge' in all cases.

22/20. Change 'and' to 'of'

22/31. It is very odd to say " 'observations' " – the reader is left to wonder about these quote unquote 'observations'. Instead, give the more detailed information on what is used for the comparison.

23/25-32. Rather than just say at the end of the paragraph that it 'is also more consistent with the focus on populations of glaciers rather than individual glaciers in this study', it would be appropriate to state clearly mid-section that the parameterization is not appropriate for use in projecting individual glacier behavior.

24/15+. The concern and fix included here re: using flux-weighted means should be included in the main text.

Figure 1. Nowhere in the paper do you discuss the implications for using different models for past forcing v. future forcing. You note that the time series are aligned through bias adjustment, but I'd like to know more about the effect of using one set of data for creating the parameterization and another set for projecting. Given that the parameterization will likely be used in a wide set of models using a variety of climate forcing data sources, does use of a single set for establishing the parameterization create any limits in confidence?

Figure 3. Include in parentheses the data source for the bathymetry. Also change 'TF' to '(TF)' in first line of caption.

Figure 6. There is no longer an (a), (b), and (c) panel. Caption needs correction.

Figure S4. Make this into two rows of three and increase the size so that it's easier to see.

Figure S5. Suggest removing panel (c).

---

## Referee Comment (RC2) · Ellyn Enderlin (Referee) · 31 May 2019

Summary: This paper pairs observations of terminus position change from 191 glaciers around Greenland with runoff and ocean temperature model outputs to devise a simple parameterization that relates terminus position change and submarine melting. Through this analysis, they find that 23% of terminus position change can be explained by variations in submarine melting (at the ice sheet scale). The authors then use the distribution of submarine melt coefficients and projected runoff and ocean temperature change to construct terminus change distributions for the ice sheet through the 21st century. Using this approach, the median terminus retreat driven by submarine melting is near zero under RCP2.6 forcing but the median retreat jumps to 5.8km, with >20km of retreat for 19% of glaciers, under RCP8.5. The paper is very well written and well

motivated, with fairly minor changes recommended to ensure that the methods and interpretation of results are clear. Major Comments: 1. The description of the runoff estimates from RACMO would benefit from a little bit more detail. When you state that you "consider the mean summer runoff over June, July and August", do you mean that you calculate that mean and apply it uniformly throughout the summer (those same three months) when calibrating the submarine melt parameterization? Or do you calculate the mean and apply a normal distribution to spread it across the summer when calibrating the submarine melt parameterization? Essentially it's unclear what you mean when you say that you "consider" these data and I think the addition of a few more sentences explaining the method would be helpful. As a follow-up, if you use summer data here with the argument that most terminus retreat occurs in summer, then why use annual data to characterize ocean thermal forcing? 2. At the top of P16 you describe how you come up with your projections using the K values constructed from observations. Since your projections are very strongly dependent on the K values, this explanation needs to be expanded so that the reader can follow precisely what you have done to devise the projections. It's not clear why you sample the K coefficient 10^4 times from the distribution. Are you extracting K values at random from the distribution for each year for each glacier? Do you extract the K values at random for each year but apply those same values to all glaciers? Are the values extracted over even shorter time periods (say every month) since the magnitude of retreat can vary within a melt season? These are important details that need to be included in the text.

3. I agree with the first reviewer that great care must be taken when presenting the results. The authors note that only a small percentage of terminus retreat can be explained by submarine melt change at the ice sheet scale and reiterate that submarine melting is only one control on terminus position in the discussion, but I agree with the first reviewer that the presentation of a handful of glaciers for demonstration purposes (focusing on Figure 10) may be misleading to someone who does not read the paper thoroughly. Minor Comments: P1,L8: "terminus position change" P1,L10: Insert comma after "considered" P2,L1: I don't think you really describe the chain of

events that lead to submarine melting and/or calving and the associated ice dynamics response. Replace "the described chain of" with something more like "the critical" or "the fundamental". P2,L4-6: I don't think you need anything about floating tongues here because then the rest of the paragraph reads like you are talking about processes occurring just where there is floating ice. If you want to call out that only a handful of glaciers outside of the far north have floating termini in regard to the modeling requirements/complexities, that can be done later on where appropriate. P2,L21-22 and P3,L1-4: Just a personal preference and you can leave it as is, but I think it would be beneficial to take these sentences (summarizing that it's challenging to model ocean and glacier processing right at the termini) to the beginning of the paragraph starting P3,L5 or even remove them altogether. P4,L29: Why 5 years? What if you looked at decadal data instead? Are the results influenced by this time scale? P5: I like the schematic in Figure 1. I think it's a good simple visualization of the approach and datasets. P5,L8: How are they significant? The biggest contributors to discharge? Fastest flowing? I recommend replacing with "fastest flowing" since you mention that 191/211 of the fastest flowing glaciers are included later in the paragraph. P9,L3: You presumably mean bias between the runoff and thermal forcing estimates from 1960-present and the respective MIROC5 data for overlaps in space and time, correct? Instead of "bias corrected", it would be clearer to say that the MIROC5 datasets were adjusted to eliminate systematic offsets with runoff from RACMO and thermal forcing from EN4. Right now this is relegated to an appendix (Appendix A) but I think the contents of this short appendix could easily be merged into the main text without bogging-down the reader. P9,L10: Like my previous comment, it would be helpful to include slightly more detail here. You remove linear trends over time from the data then fit a linear trendline to the detrended submarine melt rate (independent variable) and detrended terminus position (dependent variable) data, correct? It gets a little confusing in here with the multiple mentions of linear trend fitting. A supplemental figure showing the example time series with their linear trends and a scatterplot showing the detrended data and the linear regression applied to assess correlation would be help-

ful. Figure 7 caption: There are two sentences in here that I feel should be moved to the text since they contain information relevant to the analysis. "Anomalies are calculated per glacier as the difference from the mean over the full timeseries available for each glacier." should be moved to the text on P11 where anomalies are mentioned but there is no description of how the anomalies are calculated in the text. The last sentence of the caption should also be moved to the text so that the definition of significance (described on P13) is clear. Figure 10 caption: You can really trim this down. Right now it reads like another paragraph in the text when I think you only need to explain (1) what scenario these are for, (2) what the lines and shading mean, and (3) that the projections are smoothed. Additional discussion point: It is worth mentioning somewhere that although you deliberately excluded some glaciers with perennial floating tongues, there were many glaciers that historically terminating as floating ice that have become grounded in recent years. Jakobshavn is the most well documented example. The break-up of floating tongues may result in an apparent decrease in the relationship between terminus position change and submarine melting since the grounded ice may retreat more slowly than the tongue collapse due to differences in ice dynamics and submarine melting with the evolution of the glacier geometry.

---

## Author Comment (AC1) · 26 Jul 2019

We thank the reviewers for their thoughtful and constructive reviews and are pleased that the manuscript was well received. Please find attached our response to reviews, revised manuscript and supporting information, and a version of the manuscript with changes marked.

Please also note the supplement to this comment:
https://www.the-cryosphere-discuss.net/tc-2019-98/tc-2019-98-AC1-supplement.zip

---

## Author Response (AR1)

*Response to reviews*

*We thank the reviewers for their thoughtful and constructive reviews and are pleased that the manuscript was well received.*

*Both reviewers had concerns regarding our presentation of projections for individual glaciers, when it is clear from the manuscript that the parameterisation is best for groups of glaciers. To address these concerns we have removed the former Figure 10 and all discussion of individual projections except that which is required for clarity or comparison to previous work.*

*We have also endeavored to make clear the limitations of the parameterisation in each relevant part of the manuscript, and we thank the reviewers for noting that we already tried to write the manuscript with this philosophy in mind. Similarly, as suggested by Reviewer 1 we have changed the title and feel that the new title – 'Estimating Greenland tidewater glacier retreat driven by submarine melting' – more appropriately represents the study.*

*Reviewer 2 requested more detail on processing of inputs and projections and these details have been added. Lastly we have also addressed each of the minor reviewer comments. Our specific responses are the italic blue paragraphs below. Where line numbers are given, they refer to the version of the manuscript with changes marked on.*

**Reviewer 1**

This paper presents a new parameterization for Greenland marine-terminating glacier retreat based on subglacial runoff and ocean thermal forcing. The authors use data from 1960-present to develop and test the parameterization, and model projections to 2100 to examine the parameterization projections to 2100. Overall, the paper is well written and easy to follow. Data and methods are explained in appropriate detail with useful and clear figures. Apart from the comments below the paper is thorough in discussing the assumptions, limits, opportunities, and other research related to this contribution. The paper will be a valuable reference for ice sheet modelers and the cryosphere community, particularly given that it outlines the recommended parameterization method for ISMIP6. It also provides an excellent discussion of the methods and opportunities for improvement, and the paper will be an important launching point for further research. Progress on incorporating ice sheet-ocean interaction into ice sheet models is critical, and the step provided in this paper is welcome.

I do have some concerns about discussion of projections made in the paper and think that adjustments are needed in the discussion and messaging to ensure that the authors are not overstepping the limits of the science. Throughout the paper the authors emphasize that there is a statistically significant relationship between terminus change and submarine melt (here $Q^{0.4}$TF), but Figure 7b also highlights how tenuous this relationship is, and the authors results themselves also demonstrate this (e.g., how much of terminus change is explained by submarine melt, etc.). It is clear that the authors are well aware of this, as they emphasize multiple times that the parameterization is best for groups of glaciers (or even an ice-sheet-side grouping), and that there are other components of the system, like topography, that may play a similar or larger role in determining retreat. Despite these statements, however, the authors still move forward on including projections for individual glaciers as well as ice-sheet-wide projections regarding retreat. These items are misleading and likely to cause confusion and misuse by readers who are not as familiar with the details of marine-terminating systems or are coming to the paper specifically for this projected information but with little attention to the paper details. By including these details (inappropriately I suggest), the authors are providing projections that can too easily be taken out of

context or cited without inclusion of the nuances and caveats that appear in this manuscript. My feeling is that because there is a statistically significant relationship, the authors have in several instances pushed their statements beyond what is actually justified by the data (and especially considering that the parameterization does not account for other major known influences, like topography). There are two areas that are particularly problematic:

*We absolutely agree that the limitations of the parameterisation must always be borne in mind, and this was very prominent in our thoughts when drafting the manuscript. We also however agree that the instances noted by the reviewer may have strayed beyond this philosophy and so we have made all changes suggested as described below.*

1) The title. The main point of this paper is a new parameterization. The paper is not providing new and improved details on past glacier response to submarine melting nor future projections with a confidence level that warrants the current title. A more appropriate title would be something along the lines of: 'New parameterization for incorporating Greenland tidewater glacier response to submarine melt in ice sheet models' or 'Parameterizing Greenland tidewater glacier response to submarine melting for improved ice sheet projections'

*While we agree the main point of the paper is the parameterisation, we also do think there are points of scientific interest relating to the interpretation of historical tidewater glacier retreat. For example, to our knowledge, this study is the first to have examined the link between tidewater glacier retreat and submarine melting over the full ice sheet and over multidecadal timescales. This allows us to examine quantitatively the extent to which glaciers respond similarly/differently to submarine melt forcing (e.g. the spread in k values in Figure 5). Therefore we hope the manuscript will be read not only by ice sheet modelers but by the wider community studying the interaction between the ice sheet and ocean. For this reason we are not keen to include 'parameterisation' or 'ice sheet models' in the title, and hope that our suggested new title 'Estimating Greenland tidewater glacier retreat driven by submarine melting' might be considered a compromise.*

2) Figure 10 and related discussion of projections. First, labeling on Figure 10 is misleading. 'Helheim-like' (and similar) has little functional meaning – like Helheim in terms of climate, bed topography, shape, ice flux, etc.? While the caption points out that these are projections that are more appropriately commenting on regional glacier behavior, the figure and elements of the discussion (e.g., 16/15 specifically states 'retreat for Helheim…') do not reflect this. Second, given the limits to the technique, I find little justification for discussing projections of individual glaciers – or even of ice-sheet-wide glaciers - in any context. Using this parameterization alone to project future glacier behavior (e.g, 17/15) is problematic.

*We included the individual glacier projections to illustrate the process that leads to the final projections: we first project future climate forcing (TF per sector, runoff per glacier), then combine these to project retreat at individual glaciers, and then finally group projections by sector. We accept however that the plotting of individual projections opens these projections up to misinterpretation and so we have removed the former Figure 10 and all discussion of individual glacier projections, except that which is necessary to explain what we have done (e.g. P16L15+). We have also retained the discussion of individual glacier projections in the comparison to previous work (Nick et al. 2013 and Beckman et al. 2018) because we do think it is interesting and important to contrast our approach with their process-based models (section 4.3).*

*We do think it is justifiable to show ice sheet-wide and sector retreat projections since Fig. 8 shows that the parameterisation is able to successfully capture past retreat for groups of glaciers. Furthermore these sector retreat projections will be implemented by ice sheet models taking part in*

*ISMIP6 and so it is important to display the magnitude and spatial variability of projected retreat. As such we have added time series of projected sector retreat to the former Fig. 11 (now Fig. 10).*

Also, the broader issue of overstating the conclusions in ways that do not reflect the full limits of the parameterization does rear its head in a few other places, and I have tried to capture those in the specific comments below. Once these elements of the discussion align appropriately with the actual skill (and known limits) of the parameterization, this paper will be an excellent addition to The Cryosphere.

*Thank you. We have addressed the specific comments below.*

One other general comment – I would like to see some discussion regarding the assumption that surface topography is unchanged. This assumption is made for developing the parameterization from past data, and in projecting future changes. Substantial peripheral thinning across Greenland is well documented, however, and increasing future runoff suggests that this trend will continue or accelerate. Ice thickness is an important component for retreat, and I am concerned that without considering it in future projections, the authors are omitting an important system change, but it is not as well acknowledged or discussed as some other omitted factors.

*Agreed, this is an important point. In terms of the peripheral thinning across Greenland, one could partition this into dynamic thinning and thinning due to surface mass balance. Near the fronts of tidewater glaciers over the past few decades, dynamic thinning has dominated over surface mass balance thinning (Csatho et al. 2014; Bevan et al., 2015). Because dynamic thinning is furthermore a response to terminus retreat, then this dynamic thinning would be captured by an ice sheet model employing our retreat parameterisation. If this dynamic thinning results in further retreat (through a positive feedback), or if the thinning is driven by surface mass balance or enhanced basal lubrication then the reviewer is correct that this is not explicitly accounted for by our parameterisation. It is possible it is implicitly accounted for because we have tuned the parameterisation to match observed retreat, and so if this positive feedback has contributed to past observed retreat, it will have affected our values of k, but we agree that an explicit treatment could be an important future improvement. We have added this discussion to the manuscript as suggested (P21L19+).*

**Specific comments (by page/line number)**

4/29. Specify if you are using the mean, weighted mean, etc. over the 5-year bins.

*Changed as suggested – we use the mean (P4L29).*

8/18. Label Kangerdlugssuaq in one of the figures.

*Added as suggested (Fig. 3a).*

8/24+. It is a bit confusing here to start by saying runoff and TF are estimated using MIROC5, but then explain that runoff is actually from MAR forced by MIROC5. Better to simplify and streamline here by going straight for – runoff from MAR forced by MIROC5 and TF from MIROC5.

*Changed as suggested (P9L4).*

11/14. Changing 'but' to 'except' makes this sentence easier to understand.

*Changed as suggested (P13L1).*

15/17. Consider including plots related to the RCP2.6 scenario in the supplementary material. These can be valuable for communicating the substantial differences in retreat under an RCP2.6 v. RCP8.5 scenario.

*Agreed - added to the supplementary material as suggested (P16L14 and Fig. S8).*

13/30. In essence, this parameterization assumes that the *setting* of a region of glaciers – and by this I mean all of the other elements influencing behavior that are outside of submarine melt – will remain the same in the future as it has been in the past. This is what provides regional or ice sheet scale confidence for using the parameterization in projections. I think it would be appropriate for the authors to more clearly acknowledge this. Given the importance of topography for short-term AND long-term retreat behavior, it is important to raise this issue as intrinsically tied to the assumptions made regarding future terminus change.

*Agreed - we are assuming that the setting of a region of glaciers does not change – i.e. that they respond to submarine melting in the future as they have in the past. Of course, this will not be the case if the glacier retreats onto land, and we have now 'masked' our projections for retreat onto land, ensuring that a glacier cannot retreat further than the position at which it becomes land-terminating (new Fig. S9 and P17L2). This has resulted in small changes to some of the numbers in the abstract, discussion and conclusion, for example the median projected retreat is now 4.2 km (formerly 5.8 km) and 27% of glaciers are projected to retreat by more than 10 km (formerly 35%).*

*Beyond this, we might also expect glaciers to retreat more slowly if they retreat into shallower water (but not completely onto land), and this is not taken into account by the current parameterisation. Another way in which glaciers might respond differently in the future compared to the past would be if their subglacial hydrology changes, which could influence basal lubrication and the distribution of plumes at their calving fronts, or if a region of glaciers retreats into an area of bedrock that is characteristically rougher than before. Since none of these factors are taken into account by the parameterisation, we are as the reviewer says essentially assuming they do not change. We have added these caveats and discussion to the manuscript (P18L14+).*

17/11. Given the parameterization, 'similar-sized glaciers' is more correctly 'similar-sized glacier basins'.

*Following our removal of Fig. 10, this sentence has been removed, but we have added this clarification to new sentences on P17L25.*

18/11. While bed topography does control short-term variability, it is also unquestionably a major factor in determining long-term change when it comes to patterns of retreat. For example, the multi-centurial behavior a glacier that extends into a long overdeepening (e.g., Jakobshavn) will be dramatically different than a glacier that may retreat onto shallow topography or even become land-terminating.

*Agreed. We have now masked our retreat projections so that the parameterisation is no longer applied once a glacier becomes land-terminating. Specifically, for each glacier we use BedMachinev3 to calculate a required retreat for the glacier to become land-terminating (new Fig. S9) and cap projected retreat at this value (P17L2). This has resulted in small changes to the distributions in Fig. 10 and the numbers in the abstract, discussion and conclusion. We thank the reviewer for bringing this up, though we note that this effect is already being taken into account in the ISMIP6 simulations because the ice sheet models do not apply the retreat parameterisation once ice retreats onto land.*

*In this way, the long-term retreat of Jakobshavn would be very different from a smaller glacier that retreats quickly onto land. See also our response to comment on line 13/30 above.*

18/32. Thinning may also occur directly at the terminus from SMB changes.

*Yes – we have now added this point to the discussion on ice sheet peripheral thinning and the potential impact on terminus position (P21L19+).*

19/3-5. This sentence is overly confident. Perhaps something like: 'The principal advantages of the parameterization are its simplicity and context provided via empirical validation, thus the critical interaction of the ice sheet with the ocean can be represented in a manner which is informed by observations and scales to large region and ice-sheet-wide applications.'

*We have revised the sentence as suggested (P19L8+).*

21/10. Delete 'behind'

*We feel that it is important here to emphasize that the specification of terminus position, plus their ice flow model, allowed them to capture dynamic mass loss of the whole glacier. Thus we feel 'behind' was important here, but have changed it to 'upstream' (P21L17).*

21/12 & 17, 22/3. Consider removing 'huge' in all cases.

*Removed as suggested (P21L28 & 33, P22L16).*

22/20. Change 'and' to 'of'

*Thank you for spotting this – changed as suggested (P22L33).*

22/31. It is very odd to say " 'observations' " – the reader is left to wonder about these quote unquote 'observations'. Instead, give the more detailed information on what is used for the comparison.

*Agreed – we have reworded 'observations' as 'best estimates' and are now more explicit about what is used to do the comparison (P23L18 and Appendix A more generally).*

23/25-32. Rather than just say at the end of the paragraph that it 'is also more consistent with the focus on populations of glaciers rather than individual glaciers in this study', it would be appropriate to state clearly mid-section that the parameterization is not appropriate for use in projecting individual glacier behavior.

*We have in fact now removed Appendix B (see response to next comment) and so this passage is no longer in the paper.*

24/15+. The concern and fix included here re: using flux-weighted means should be included in the main text.

*We absolutely agree that this concern and fix is important when applying sector-averaged retreat to an ice sheet model. However, we want the focus of this paper to be on motivating, calibrating and validating the retreat parameterisation, and providing a brief projection as an illustration, rather than on the technicalities of applying the retreat parameterisation to an ice sheet model.*

*Furthermore, since we submitted this paper two months ago, it has also become clear that there will need to be a companion Greenland ocean forcing ISMIP6 paper (Slater et al., in prep.). The second paper will deal with the specifics of applying the retreat parameterisation to ice sheet models, and will additionally cover many other aspects of the ISMIP6 project, including for example the selection of CMIP5 models. We think that the content of this appendix would be better situated in that second paper, and are therefore proposing removing this appendix from the current paper. We note that the appendix was referred to only very briefly in the main text, and is not at all central to the results or discussion so that removing this appendix does not diminish the paper. We hope the reviewer will understand this proposition.*

Figure 1. Nowhere in the paper do you discuss the implications for using different models for past forcing v. future forcing. You note that the time series are aligned through bias adjustment, but I'd like to know more about the effect of using one set of data for creating the parameterization and another set for projecting. Given that the parameterization will likely be used in a wide set of models using a variety of climate forcing data sources, does use of a single set for establishing the parameterization create any limits in confidence?

*It is unfortunately a necessity to use different models for past vs future forcing because within the ISMIP6 process, the CMIP models (of which MIROC5 is one) that are used to form future forcing do not represent past climate variability well, and therefore cannot be used for establishing the parameterisation. With regard specifically to the surface runoff, we used RACMO2.3p2 forced by ERA-Interim for the past because it is the highest resolution regional climate model at present (Noel et al., 2018), and we used MAR3.9.6 for the future because it has a history of running future projections (Fettweis et al., 2013) and is the regional climate model that will be used for surface mass balance projections in the ISMIP6 process. We believe this RACMO vs MAR model mismatch is however not a problem because the bias adjustment between the two is small (that is, the bias adjustment is typically smaller than the interannual variability in either model). The ocean bias corrections are much more significant (being larger than the interannual variability in the EN4 dataset or CMIP models). Therefore is it unavoidable to use different models for past and future, and some of the bias corrections are large. We have made this clear in the revised manuscript by adding discussion and an additional figure and table to the supplement (Appendix A, Fig. S13, Table S2).*

*In the analysis leading to this paper we did consider alternative forcing datasets for establishing the parameterisation. From the ocean side, there are global ocean climatologies such as the World Ocean Atlas 2013 and 2018 (Locarnini et al., 2013; 2018) and regional ocean climatologies such as the Greenland, Iceland and Norwegian Seas Climatology (Seidov et al., 2018) and the Arctic Regional Climatology (Boyer et al., 2012). As might be expected from the fact that all these climatologies use much of the same oceanographic data, we did not find significant differences between these climatologies and EN4 (e.g. Fig. R1). Ultimately we preferred EN4 as it covers the full ocean area required (in contrast to the regional climatologies) and is available at annual resolution (in contrast to the World Ocean Atlas that only provides decadal fields). We also considered ocean reanalyses such as CHORE (Yang et al. 2016) that are not independent from the ocean climatologies since the ocean reanalyses typically assimilate the ocean climatologies to improve their match with observations. Again we did not find substantial differences relative to the climatologies.*

*Similarly we also considered two runoff products for the past – the RACMO2.3p2 simulation used in the paper and a longer-term 1900-2010 MAR3.5.2 simulation (Fettweis et al. 2017). Once more we did not find substantial differences between these two runoff estimates (e.g. Fig. R2), and we prioritized the RACMO dataset due to its 5.5 km resolution that is important in resolving the surface mass balance of narrow tidewater glaciers (Noel et al. 2018).*

[Figure]

*Fig. R1: Difference between the EN4 and WOA13 and WOA18 ocean climatologies, for the time period 1985-present in which most of our terminus position records are found. (a) shows the time-mean 200-500 m ocean temperature in WOA18. A time series for each of the 4 boxes is shown in (b)-(e); these 4 boxes are different from the 7 sectors considered in the main paper but are still suitable for comparing the climatologies. Each of (b)-(e) shows the annual EN4 time series (red), and the mean EN4 over decadal time periods (horizontal red) for comparison to the WOA13 (dashed blue) and WOA18 (dashed black) datasets that are only available on these decadal time periods. Both the absolute value of temperature and the temporal evolution are similar in EN4 and WOA13 or WOA18.*

[Figure]

*Fig. R2: RACMO2.3p2 vs MAR3.5.2 runoff estimates for Jakobshavn (left) and for all 191 glaciers (right).*

*Given all of these methods of estimating past forcing agree relatively well, we do not think using an additional calibration dataset would make much difference to the parameterisation, and therefore the use of a single set of forcing data for establishing the parameterisation does not diminish our confidence in the parameterisation. We do think that the calibration of the parameterisation could be improved for example by accounting for the uncertainties in the forcing data, perhaps in a Monte-Carlo fashion, but at this point we would leave this to future work. The key points from this discussion have now been added to the paper (P6L7, P8L3, P9L10).*

Figure 3. Include in parentheses the data source for the bathymetry. Also change 'TF' to '(TF)' in first line of caption.

*Source added. Brackets added to TF (Fig. 3).*

Figure 6. There is no longer an (a), (b), and (c) panel. Caption needs correction.

*Corrected – thank you for spotting this (Fig. 6).*

Figure S4. Make this into two rows of three and increase the size so that it's easier to see.

*Changed as suggested (now Fig. S5).*

Figure S5. Suggest removing panel (c).

*Removed as suggested (now Fig. S10).*

**Reviewer 2: Ellyn Enderlin**

Summary: This paper pairs observations of terminus position change from 191 glaciers around Greenland with runoff and ocean temperature model outputs to devise a simple parameterization that relates terminus position change and submarine melting. Through this analysis, they find that 23% of terminus position change can be explained by variations in submarine melting (at the ice sheet scale). The authors then use the distribution of submarine melt coefficients and projected runoff and ocean temperature change to construct terminus change distributions for the ice sheet through the 21st century. Using this approach, the median terminus retreat driven by submarine melting is near zero under RCP2.6 forcing but the median retreat jumps to 5.8km, with >20km of retreat for 19% of glaciers, under RCP8.5. The paper is very well written and well-motivated, with fairly minor changes recommended to ensure that the methods and interpretation of results are clear.

**Major Comments:**

1. The description of the runoff estimates from RACMO would benefit from a little bit more detail. When you state that you "consider the mean summer runoff over June, July and August", do you mean that you calculate that mean and apply it uniformly throughout the summer (those same three months) when calibrating the submarine melt parameterization? Or do you calculate the mean and apply a normal distribution to spread it across the summer when calibrating the submarine melt parameterization? Essentially it's unclear what you mean when you say that you "consider" these data and I think the addition of a few more sentences explaining the method would be helpful. As a

follow-up, if you use summer data here with the argument that most terminus retreat occurs in summer, then why use annual data to characterize ocean thermal forcing?

*Thank you for suggesting these clarifications. At no point in the analysis do we consider timescales shorter than annual, and so we have not made any assumption of how the runoff is spread throughout the year. We simply consider 1 value of runoff per glacier per year, which is the June-July-August mean. Ultimately when calibrating the retreat parameterisation, we also use 5-year bins, and so the number which enters the calibration is the mean over 5 years of the JJA mean runoff in each year. We have added this clarification to the manuscript (P7L2 and P7L4).*

*We admit that the use of JJA runoff but annual ocean thermal forcing is a little inconsistent, and in fact this was brought up as a concern by one of the co-authors during drafting of the manuscript. Since oceanographic data is in general sparse (e.g. Figs. S4 & S5), we were unconvinced that we could resolve a seasonal cycle with confidence, and we therefore considered the annual means to be more robust. Then the question is why not use annual mean runoff values? The decision to use JJA runoff rather than annual values was motivated partly by the fact that past submarine melt rate parameterisations (e.g. Slater et al., 2016; Rignot et al., 2016) have typically been calibrated to summer runoff values, and partly by the fact that we find JJA runoff values more intuitive. A new plot added to the supporting information (Fig. S3) shows that there is a very close relationship between annual and JJA runoff, whereby Q_annual = 0.26\*Q_JJA in both the past and projection. Therefore the use of annual instead of JJA values to calibrate the retreat parameterisation would make very little difference (the k-values would be a factor of 1/0.26 = 3.8 larger, but this would be completely compensated by the use of annual runoff in the projections, which is a factor 3.8 smaller). This point is now clarified in the text (P7L5 and new Fig. S3).*

2. At the top of P16 you describe how you come up with your projections using the K values constructed from observations. Since your projections are very strongly dependent on the K values, this explanation needs to be expanded so that the reader can follow precisely what you have done to devise the projections. It's not clear why you sample the K coefficient $10^4$ times from the distribution. Are you extracting K values at random from the distribution for each year for each glacier? Do you extract the K values at random for each year but apply those same values to all glaciers? Are the values extracted over even shorter time periods (say every month) since the magnitude of retreat can vary within a melt season? These are important details that need to be included in the text.

*Thank you for this suggestion – you are right that this is critical and so we have expanded this description. We randomly sample from the distribution so that, in the projections, we capture the past diversity of glacier response to change in submarine melting (section 3.2). For each glacier, we extract a K value at random from the distribution shown in Fig. 5a, call this value K0. This K value, K0, is assumed to be constant throughout the time period of the projection. We project retreat as K0\*F(t), where F(t) is the smoothed $Q^{0.4}TF$ time series for the glacier (e.g. Fig. 9c). Again, we do not consider any time periods shorter than 1 year – the time series of $Q^{0.4}TF$ is constructed at annual resolution and is then smoothed to F(t) so the true resolution is even longer than annual. We repeat this procedure $10^4$ times, giving $10^4$ retreat projections K0\*F(t) for each glacier. We did not consider letting K vary in time – this is an interesting idea, but keeping it constant in time is more consistent with how we have calibrated the parameterisation, in which we calculated a single value of K for the whole time period 1960-present. We have added these clarifications to the manuscript (P16L15+).*

3. I agree with the first reviewer that great care must be taken when presenting the results. The authors note that only a small percentage of terminus retreat can be explained by submarine melt

change at the ice sheet scale and reiterate that submarine melting is only one control on terminus position in the discussion, but I agree with the first reviewer that the presentation of a handful of glaciers for demonstration purposes (focusing on Figure 10) may be misleading to someone who does not read the paper thoroughly.

*Agreed – as described above in the summary and response to reviewer 1, we initially included these individual glacier projections as we thought they would be a useful illustration of the process that leads to the sector-by-sector projections (it is necessary to project retreat at individual glaciers first and then zoom out to the sectors). After this review process it seems these individual projections could be taken out of context and so we have now removed the former Fig. 10 and all discussion of individual projections except that which is necessary to explain how we get to the sector-level projections.*

**Minor Comments:**

P1,L8: "terminus position change"

*Changed as suggested (P1L8).*

P1,L10: Insert comma after "considered"

*Changed as suggested (P1L10).*

P2,L1: I don't think you really describe the chain of events that lead to submarine melting and/or calving and the associated ice dynamics response. Replace "the described chain of" with something more like "the critical" or "the fundamental".

*Changed as suggested (P2L1).*

P2,L4-6: I don't think you need anything about floating tongues here because then the rest of the paragraph reads like you are talking about processes occurring just where there is floating ice. If you want to call out that only a handful of glaciers outside of the far north have floating termini in regard to the modeling requirements/complexities, that can be done later on where appropriate.

*Agreed, this sentence has been removed (P2L4-6).*

P2,L21-22 and P3,L1-4: Just a personal preference and you can leave it as is, but I think it would be beneficial to take these sentences (summarizing that it's challenging to model ocean and glacier processing right at the termini) to the beginning of the paragraph starting P3,L5 or even remove them altogether.

*Changed as suggested – we think these sentences are a little redundant and so have removed them altogether (except the former P3L2-4, which states that we lack a validated calving law).*

P4,L29: Why 5 years? What if you looked at decadal data instead? Are the results influenced by this time scale?

*For binning on timescales longer than 5 years, the length of the data record becomes an issue. Many of our terminus position records are 20-30 years in length (Fig. 5d), which is 4-6 data points with 5-year bins but only 2-3 data points with decadal bins. Thus when fitting trends in time you work with fewer data points with longer bins. Equally we felt that timescales shorter than 5 years could be*

*difficult to capture with such a simple parameterisation, and we are really interested in the longer multi-decadal terminus position and sea level trend in ISMIP6. Thus 5 years was chosen as a compromise. We have added a figure to supporting information (Fig. S7) showing the sensitivity to binning over different time periods, and the sensitivity to adding offsets to the time period (i.e. with 5 year binning you could choose 1990 → 1995, 1995 → 2000 etc. or 1991 → 1996, 1996 → 2001 etc.). There is only little sensitivity to these choices, arising from which bin the irregularly-spaced data falls into, but the choice made in the main paper falls centrally in the observed range, and so we feel our choice is justified and the results are not significantly affected by this choice. This statement and a reference to the new supporting information figure have been added to the manuscript (P4L31 and Fig. S7).*

P5: I like the schematic in Figure 1. I think it's a good simple visualization of the approach and datasets.

*Thank you.*

P5,L8: How are they significant? The biggest contributors to discharge? Fastest flowing? I recommend replacing with "fastest flowing" since you mention that 191/211 of the fastest flowing glaciers are included later in the paragraph.

*By significant we meant fastest flowing – this has been clarified (P5L8).*

P9,L3: You presumably mean bias between the runoff and thermal forcing estimates from 1960-present and the respective MIROC5 data for overlaps in space and time, correct? Instead of "bias corrected", it would be clearer to say that the MIROC5 datasets were adjusted to eliminate systematic offsets with runoff from RACMO and thermal forcing from EN4. Right now this is relegated to an appendix (Appendix A) but I think the contents of this short appendix could easily be merged into the main text without bogging-down the reader.

*We have added this clarification to the text (P9L23). It might be worth noting here that we will soon submit another paper (Slater et al., in prep) that will fully describe the ISMIP6 Greenland ocean forcing, with more detail on these bias corrections for a range of CMIP5 models. The focus of the present paper is on motivating, validating and calibrating the parameterisation, with the future projection here intended as an illustration and to show the potential of the parameterisation. As such, we would like to leave some of the technical details of the projection (such as the bias correction) to the appendix, and hope the reviewer understands this reasoning.*

P9,L10: Like my previous comment, it would be helpful to include slightly more detail here. You remove linear trends over time from the data then fit a linear trendline to the detrended submarine melt rate (independent variable) and detrended terminus position (dependent variable) data, correct? It gets a little confusing in here with the multiple mentions of linear trend fitting. A supplemental figure showing the example time series with their linear trends and a scatterplot showing the detrended data and the linear regression applied to assess correlation would be helpful.

*Thank you for this suggestion – we have added a figure to the supplement (Fig. S6) that walks the reader through this process for one of the two example glaciers shown in Fig. 4 of the main paper. In making the new supplemental figure we noticed a small coding error in the calculation of the p-values shown on Figs. 4 and 5 – this has now been corrected and has made essentially no difference to the manuscript. The new supplemental figure is signposted from the main paper (P9L29).*

Figure 7 caption: There are two sentences in here that I feel should be moved to the text since they contain information relevant to the analysis. "Anomalies are calculated per glacier as the difference from the mean over the full timeseries available for each glacier." should be moved to the text on P11 where anomalies are mentioned but there is no description of how the anomalies are calculated in the text. The last sentence of the caption should also be moved to the text so that the definition of significance (described on P13) is clear.

*We have moved these two sentences to the appropriate places in the main text (P13L4+). We have also moved a similar sentence on the definition of anomalies from the Figure 4 caption to the main text (P10L20+).*

Figure 10 caption: You can really trim this down. Right now it reads like another paragraph in the text when I think you only need to explain (1) what scenario these are for, (2) what the lines and shading mean, and (3) that the projections are smoothed. Additional discussion point: It is worth mentioning somewhere that although you deliberately excluded some glaciers with perennial floating tongues, there were many glaciers that historically terminating as floating ice that have become grounded in recent years. Jakobshavn is the most well documented example. The break-up of floating tongues may result in an apparent decrease in the relationship between terminus position change and submarine melting since the grounded ice may retreat more slowly than the tongue collapse due to differences in ice dynamics and submarine melting with the evolution of the glacier geometry.

*Following the responses above, Figure 10 has now been removed from the manuscript. We have added a sentence noting this important point about ice tongues – we agree that ice tongue break up, if treated as grounded ice retreat, could lead to larger kappa values than is appropriate for grounded retreat (P5L14+). We do not however believe that this would substantially alter our distribution of kappa (Fig. 5a) because this concern will only affect a very small proportion of glaciers.*

***Literature cited (see also references in manuscript)***

*Bevan et al. (2015), Seasonal dynamic thinning at Helheim Glacier, Earth and Planetary Science Letters, doi: 10.1016/j.epsl.2015.01.031*

*Boyer et al. (2012), Arctic Regional Climatology, Regional Climatology Team, NOAA/NODC dataset doi:10.7289/V5QC01J0*

*Fettweis et al. (2017), Reconstructions of the 1900-2015 Greenland ice sheet surface mass balance using the regional climate MAR model, The Cryosphere, doi: 10.5194/tc-11-1015-2017*

*Locarnini et al. (2018), World Ocean Atlas 2013, Volume 1: Temperature. S. Levitus, Ed., A. Mishonov Technical Ed.; NOAA Atlas NESDIS 73, 40 pp.*

*Locarnini et al. (2018), World Ocean Atlas 2018, Volume 1: Temperature. A. Mishonov Technical Ed.; in preparation.*

*Seidov et al. (2018), Greenland-Iceland-Norwegian Seas Regional Climatology version 2, Regional Climatology Team, NOAA/NCEI.*

*Slater et al. (in prep, likely submission August 2019), Ocean forcing for modeling of future sea level contribution from the Greenland ice sheet*

*Yang et al. (2016), Historical ocean reanalyses (1900-2010) using different data assimilation strategies, Quarterly Journal of the Royal Meteorological Society, doi: 10.1002/qj.2936*

---

## Author Response (AR2)

**Response to editor for tc-2019-98**

**Editor Decision: Publish subject to technical corrections**

> *We would very much like to thank both the editor and reviewers for their time, and their constructive and careful reviews of our manuscript. Given the very minor nature of the changes made we have not included a track changes document with this iteration.*

**Comments to the Author:**

Please correct the spelling of "Petermann" on p.5, l.10 and p.19, l.29.

> *Corrected – thank you for spotting this.*

Also, consider updating the dates of last access in the Data Availability section.

> *We have updated the dates where we have recently accessed these sources.*